# Golgi localized β1-adrenergic receptors stimulate Golgi PI4P hydrolysis by PLCε to regulate cardiac hypertrophy

Craig A Nash[1†], Wenhui Wei[1†], Roshanak Irannejad[2], Alan V Smrcka[1*]

[1]Department of Pharmacology, University of Michigan School of Medicine, Ann Arbor, United States; [2]Cardiovascular Research Institute, University of California, San Francisco, San Francisco, United States

**Abstract** Increased adrenergic tone resulting from cardiovascular stress leads to development of heart failure, in part, through chronic stimulation of β1 adrenergic receptors (βARs) on cardiac myocytes. Blocking these receptors is part of the basis for β-blocker therapy for heart failure. Recent data demonstrate that G protein-coupled receptors (GPCRs), including βARs, are activated intracellularly, although the biological significance is unclear. Here we investigated the functional role of Golgi βARs in rat cardiac myocytes and found they activate Golgi localized, prohypertrophic, phosphoinositide hydrolysis, that is not accessed by cell surface βAR stimulation. This pathway is accessed by the physiological neurotransmitter norepinephrine (NE) via an Oct3 organic cation transporter. Blockade of Oct3 or specific blockade of Golgi resident β1ARs prevents NE dependent cardiac myocyte hypertrophy. This clearly defines a pathway activated by internal GPCRs in a biologically relevant cell type and has implications for development of more efficacious β-blocker therapies.

DOI: https://doi.org/10.7554/eLife.48167.001

*For correspondence:
avsmrcka@umich.edu

†These authors contributed equally to this work

Competing interests: The authors declare that no competing interests exist.

## Introduction

Cardiovascular diseases, including heart failure, are the leading cause of disease and death in the developed world. In response to long term pathological stress, such as hypertension or myocardial infarction, the levels of neurohumoral factors such as epinephrine, endothelin and angiotensin II increase. These hormones directly stimulate G protein-coupled receptors (GPCRs), including β-adrenergic receptors in cardiac myocytes (*D'Angelo et al., 1997*; *Adams et al., 1998*; *Rockman et al., 2002*). Chronic stimulation of these receptors, including stimulation of β-adrenergic receptors (βARs) by catecholamines, drives cardiac hypertrophy and ventricular remodeling, ultimately leading to heart failure. The efficacy of β-blocker therapy for treatment of heart failure results, at least in part, by ameliorating chronic βAR stimulation in the heart (*Bristow, 2000*; *Wang et al., 2018*).

Previous studies by our group, and others, have implicated specific phospholipase C (PLC) enzymes downstream of Gq and Gs-coupled GPCRs in the regulation of cardiac hypertrophy. PLCβ isoforms stimulated by Gq have been implicated in cardiac hypertrophy driven by cell surface α-adrenergic receptors (αAR) (*Filtz et al., 2009*; *Grubb et al., 2011*). Our laboratory identified a critical role of PLCε in cardiac hypertrophy driven downstream of the endothelin (ET-1A) receptor and cAMP signaling by βARs (*Zhang et al., 2011*; *Zhang et al., 2013*). PLCε is poised as a nexus for multiple receptor signaling systems due to its diversity of upstream regulators including heterotrimeric G protein βγ subunits and small GTPases, including Rap, Rho and Ras, and cAMP via the Rap guanine nucleotide exchange factor (GEF), Epac (*Kelley et al., 2004*; *Malik et al., 2015*; *Kelley et al., 2001*; *Harden and Sondek, 2006*; *Smrcka et al., 2012*).

PLCε and Epac are scaffolded together at the nuclear envelope in cardiac myocytes by the hypertrophic organizer, muscle-specific A kinase anchoring protein (mAKAPβ), in close proximity to the Golgi apparatus (*Dodge-Kafka et al., 2005*). Activation of PLCε in cardiac myocytes induces hydrolysis of phosphatidylinositol-4-phosphate (PI4P) at the Golgi apparatus leading to release of diacylglycerol (DAG) and inactive $IP_2$ in the vicinity of the nucleus to facilitate nuclear protein kinase D (PKD) and subsequent downstream hypertrophic pathways (*Zhang et al., 2013*; *Vega et al., 2004*). In previous work we found that either the Epac-selective cAMP analog, cpTOME, or forskolin stimulation of adenylate cyclase, activate PLCε-dependent Golgi PI4P hydrolysis in cardiac myocytes. Surprisingly, the βAR agonist isoproterenol (Iso), does not stimulate Golgi PI4P hydrolysis despite strongly stimulating cAMP production (*Nash et al., 2018*). As an explanation for this apparent paradox we demonstrated PLCε-dependent PI4P hydrolysis can be controlled by two distinct pools of cAMP delimited by distinct PDE isoforms. One pool, limited by PDE3 acts through Epac and Rap1 to activate PLCε, while a second pool controlled by PDE2 and/or PDE9A inhibits PLCε activity through activation of PKA (*Nash et al., 2018*). cAMP generated downstream of Iso cannot access the Epac/mAKAPβ/PLCε complex unless PDE3 is specifically inhibited, partially explaining the lack of activation of Golgi PI4P hydrolysis by Iso.

A new paradigm for generating localized cAMP signals in cells has been established by a number of recent studies indicating that GPCRs in different intracellular compartments have discrete signaling outputs (*Vilardaga et al., 2014*; *Irannejad et al., 2017*; *Irannejad et al., 2013*; *Irannejad and von Zastrow, 2014*; *Stoeber et al., 2018*; *Calebiro et al., 2009*). An emergent idea is that βARs internalized into endosomes continue to stimulate cAMP accumulation resulting in cellular outcomes distinct from those generated by βARs at the cell surface. *Irannejad et al. (2017)* established that β1ARs expressed at the Golgi apparatus in HeLa cells can induce G protein activation in the Golgi when stimulated by either a cell permeant agonist, dobutamine, through passive diffusion, or the physiological hormone epinephrine, when transported across the cell membrane by the organic cation transporter, Oct3. These ideas led us to hypothesize that in cardiac myocytes, βARs resident at the Golgi apparatus could generate cAMP that has privileged access to the Epac/mAKAPβ/PLCε scaffold at the nuclear envelope/Golgi interface, thereby yielding a set of signals divergent from those generated by cell surface βARs. We demonstrate that endogenous β1ARs at the Golgi apparatus in cardiac myocytes are required to stimulate hypertrophic Epac/PLCε-dependent PI4P hydrolysis at the Golgi, and that these intracellular βARs can be accessed by physiological neurotransmitters, and synthetic β-blockers and agonists. These data present a potential new paradigm for drug development for treating heart failure through deliberate targeting of internal βARs in cardiac myocytes.

## Results

### Dobutamine induces β1-AR activation and PI4P hydrolysis at the Golgi

In previous studies we showed that cAMP produced upon stimulation with the βAR agonist, Iso, does not induce PI4P hydrolysis at the Golgi in NRVMs without addition of a PDE3 inhibitor (*Nash et al., 2018*). We hypothesized that intracellular βARs may be required to generate a specific pool of cAMP with privileged access to the Epac/mAKAPβ/PLCε complex, and that the cell impermeant agonist, Iso, is unable to access these internal receptors. To test this hypothesis, we utilized the membrane permeant β1AR-agonist, dobutamine, either alone, or in combination with the βAR antagonists, metoprolol and sotalol, which are membrane-permeant and -impermeant, respectively (*Irannejad et al., 2017*). NRVMs were transduced with an adenovirus expressing the PI4P biosensor, FAPP-PH-GFP and stimulated with dobutamine (100 nM) or PBS. In contrast to Iso, dobutamine stimulated rapid and sustained PI4P hydrolysis (*Figure 1A*, left). Addition of metoprolol, but not sotalol, blocked dobutamine-stimulated PI4P hydrolysis (*Figure 1A* middle and right panels). This indicates that stimulation of an intracellular population βARs is required for dobutamine-mediated PI4P hydrolysis. To visualize βAR activation at the Golgi apparatus, NRVMs were transfected with translocatable Venus-based sensor of Gs coupled receptor activation, NES-Venus-mini-Gs (*Wan et al., 2018*). When a Gs coupled receptor is activated, NES-Venus-mini-Gs binds to the activated receptor, which is then visualized as translocation from the cytoplasm to membranes where the activated receptor is located, providing information on receptor activation in subcellular compartments. In unstimulated cells NES-Venus-mini-Gs is distributed throughout the cytoplasm. We did not detect translocation of

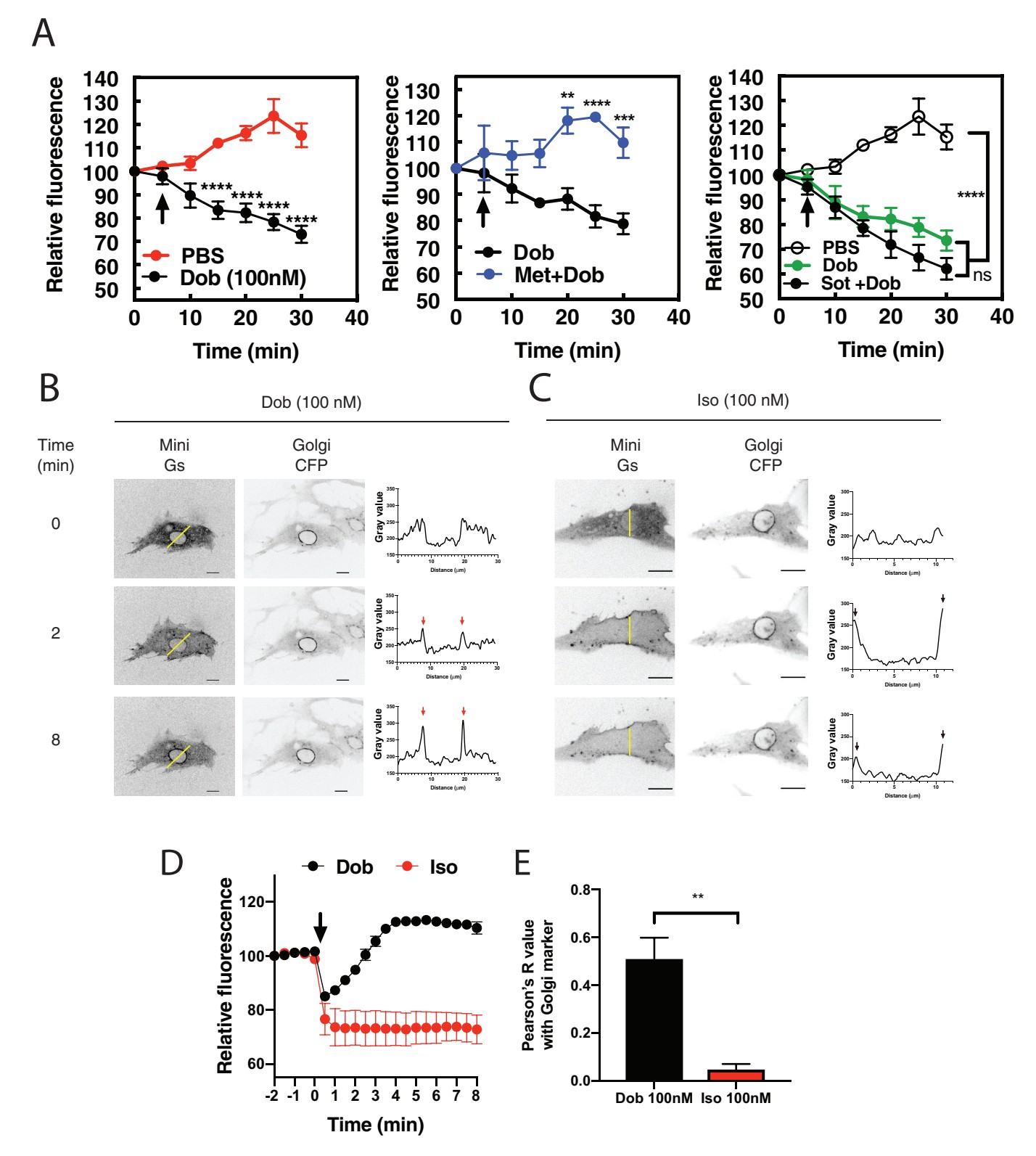

**Figure 1.** Dobutamine induces PI4P hydrolysis through the activation of internal βARs. (**A**) NRVMs were transduced with FAPP-PH-GFP and stimulated as indicated. Time lapse live cell microscopy was used to quantitate Golgi associated FAPP-PH-GFP fluorescence was quantitated as previously described (*Zhang et al., 2013*; *Malik et al., 2015*). NRVMs were stimulated with dobutamine alone (100 nM, left) (N = 9), dobutamine in the presence or absence of metoprolol (100 µM, center) (N = 4), or dobutamine in the presence or absence of sotalol (5 mM, right) (N = 8). Data are not significantly

*Figure 1 continued on next page*

*Figure 1 continued*

different between dobutamine and dobutamine + sotalol. Data are from at least 4 cells for each N. (**B**) NRVMs were transfected with β1-ARs and NES-Venus-mini-Gs, followed by viral transduction with CFP-Giantin. Representative confocal fluorescence images of dobutamine-mediated NES-Venus-mini-Gs recruitment (100 nM, B left), CFP-Giantin Golgi marker (B, center) and histogram of representative NES-Venus-mini-Gs recruitment (B, right). Red arrow = Perinuclear region, Black arrow = sarcolemma. The yellow line indicates where histogram data was captured. Scale bars are 10 μm. (**C**) Representative confocal fluorescence images of Iso-mediated NES-Venus-mini-Gs recruitment (100 nM, left), CFP-Giantin Golgi marker (B, center) and histogram of representative NES-Venus-mini-Gs recruitment. Yellow line indicates where histogram data was captured. Scale bars = 10 μm (**D**) Mean data of fluorescence intensity of NES-Venus-mini-Gs at the perinuclear region corresponding to the Golgi ± SEM from at least 5 cells. (**E**) Pearson's correlation coefficient for overlap of YFP-Mini-Gs and CFP-Giantin images. All symbols on time course graphs are presented as mean ± standard error from N = 4 or more independent preparations of myocytes. Agonists were added where indicated by the arrow.

DOI: https://doi.org/10.7554/eLife.48167.002

The following video, source data, and figure supplements are available for figure 1:

**Source data 1.** PI4P hydrolysis is stimulated by dobutamine and inhibited by a cell permeable antagonist.
DOI: https://doi.org/10.7554/eLife.48167.005
**Source data 2.** Dobutamine but not Iso stimulates Mini Gs translocation.
DOI: https://doi.org/10.7554/eLife.48167.006
**Figure supplement 1.** Staining of endogenous β1AR in cardiac myocytes.
DOI: https://doi.org/10.7554/eLife.48167.003
**Figure supplement 2.** Disruption of the Golgi apparatus reverses mini-Gs protein recruitment to the perinuclear region by dobutamine.
DOI: https://doi.org/10.7554/eLife.48167.004
**Figure 1—video 1.** Dobutamine activates β1ARs in the Golgi apparatus.
DOI: https://doi.org/10.7554/eLife.48167.007
**Figure 1—video 2.** Isoproterenol does not activate β1ARs in the Golgi apparatus.
DOI: https://doi.org/10.7554/eLife.48167.008
**Figure 1—video 3.** Disruption of the Golgi apparatus inhibits activation of β1ARs in the Golgi apparatus.
DOI: https://doi.org/10.7554/eLife.48167.009

NES-Venus-mini-Gs to endogenously expressed receptors (data not shown), although, as has been previously reported, we were able to detect staining of endogenous β1ARs at the Golgi in cardiac myocytes (*Figure 1—figure supplement 1*), so NRVMs were cotransfected with FLAG-β1AR. Addition of dobutamine to cells co-expressing NES-Venus-mini-Gs and FLAG-β1AR, caused a rapid clearance of cytoplasmic fluorescence and accumulated fluorescence at the PM and punctate structures, possibly corresponding to microtubules. This was followed by slower translocation of Venus associated fluorescence to the perinuclear region of the cell colocalizing with the CFP-giantin, a marker for the Golgi apparatus (*Figure 1B,D and E* and *Figure 1—video 1*). Iso also caused rapid translocation to the PM but, in contrast to dobutamine, no accumulation at the perinuclear region was observed (*Figure 1C,D and E* and *Figure 1—video 2*). To further confirm association of NES-Venus-mini-Gs with the Golgi, cells were treated with dobutamine for 8 min to cause NES-Venus-mini-Gs translocation to the perinuclear region, followed by treatment with Brefeldin A. Brefeldin A treatment significantly reversed NES-Venus-mini-Gs association with the perinuclear region, confirming Golgi localization of NES-Venus-mini-Gs (*Figure 1—figure supplement 2* and *Figure 1—video 3*). Taken together, these data support the idea that a population of βARs present at the Golgi can be activated by exogenous agonists and stimulate Gs, ultimately resulting in PLC activation and PI4P hydrolysis.

## PI4P hydrolysis by Dobutamine requires Golgi-localized PLCε and Epac

Previous data from our laboratory demonstrated that the Epac/mAKAPβ/PLCε complex is responsible for cAMP-mediated PI4P hydrolysis at the Golgi in NRVMs (*Zhang et al., 2013*; *Nash et al., 2018*). To determine if this complex is responsible for dobutamine stimulated PI4P hydrolysis we utilized adenoviral shRNA to deplete PLCε, or adenoviral transduction of the RA1 domain of PLCε, which competes for the interaction between PLCε and mAKAPβ, disrupting the mAKAPβ-PLCε complex (*Zhang et al., 2011*). PLCε (*PLCE1*) shRNA depletion of PLCε completely inhibited PI4P hydrolysis stimulated by dobutamine, while negative control (NC) scrambled shRNA had no effect (*Figure 2A*). Viral expression of the PLCε RA1 domain also inhibited PI4P hydrolysis stimulated by dobutamine (*Figure 2B*). This demonstrates that mAKAPβ- scaffolded PLCε is required for

dobutamine-mediated PI4P hydrolysis. The Epac-selective inhibitor HJC0726 inhibited (*Zhu et al., 2015*) PI4P hydrolysis stimulated by dobutamine (*Figure 2C*), however, the βγ inhibitor Gallein had no effect (*Figure 2D*). Taken together, these data indicate that Golgi PI4P hydrolysis stimulated by intracellular GPCRs requires the Epac/mAKAPβ/PLCε complex.

## Inhibition of Golgi βARs prevents PI4P hydrolysis stimulated by dobutamine

The data presented so far suggest that dobutamine acts through βARs at the Golgi apparatus to stimulate Golgi PI4P hydrolysis. To more directly demonstrate a requirement for endogenous βAR activation in the Golgi we targeted the βAR selective nanobody, Nb80, to the Golgi apparatus using the rapamycin inducible FRB-FKBP system. This approach has been previously used to inhibit βARs at the Golgi in Hela cells (*Irannejad et al., 2017*). NRVMs were transduced with virus containing CFP-Nb80-FRB and FKBP-mApple-GalT protein for Golgi targeting, along with a virus expressing GFP-FAPP-PH. Cells were selected that were positive for mApple, CFP and GFP fluorescence. Upon addition of rapamycin, translocation of CFP-Nb80-FRB to the Golgi apparatus was observed (*Figure 3A*). In cells pretreated with vehicle control, dobutamine stimulated PI4P hydrolysis. However, following addition of 1 µM Rapamycin, dobutamine-induced PI4P hydrolysis was significantly inhibited (*Figure 3B*). These data confirm that βARs expressed directly on the Golgi are required for dobutamine-mediated PI4P hydrolysis at the Golgi.

Nb80 can bind to and block both β1 and β2ARs, and while dobutamine is relatively selective for β1ARs over β2ARs, it can still potentially stimulate β2ARs. To confirm that dobutamine stimulated PI4P hydrolysis is through β1ARs, cells were treated with the cell permeant, highly selective, β1AR antagonist CGP-20172 (100 nM) followed by stimulation with dobutamine. CGP-20172 completely blocked dobutamine-stimulated PI4P hydrolysis (*Figure 3C*), indicating that the dobutamine stimulation of Golgi localized PI4P hydrolysis is through Golgi localized β1ARs.

## Norepinephrine (NE) induces PI4P hydrolysis and βAR activation at internal membranes in cardiac myocytes

Thus far we have shown that the synthetic β1AR selective agonist, dobutamine, can enter cells, activate intracellular βARs, and stimulate intracellular PLC activity which has interesting pharmacological implications but may not be relevant to physiological cardiac regulation. For this reason, we determined if the sympathetic neurotransmitter norepinephrine (NE) could stimulate PI4P hydrolysis and βAR activation at the Golgi in NRVMs. Cells transduced with adenovirus containing FAPP-PH-GFP were stimulated with NE (10 µM) at 37°C. NE caused delayed, but robust PI4P hydrolysis (*Figure 4A*, left). PI4P hydrolysis initiated by NE was blocked by metoprolol (100 µM, *Figure 4A*, center) but not sotalol (5 mM, *Figure 4A*, right) indicating that NE can stimulate internal βARs. To further determine if NE could enter cells and activate βARs at the Golgi, we monitored NES-Venus-mini-Gs recruitment to the Golgi in response to NE. NE initially stimulated NES-Venus-mini-Gs translocation to the plasma membrane as was observed for dobutamine, followed by slower, NES-Venus-mini-Gs translocation to the Golgi (*Figure 4B* and *Figure 4—video 1*). Since PM and Golgi recruitment are difficult to see in the same confocal plane for *Figure 4—video 2* a separate cell was imaged in a confocal plane that emphasizes Golgi recruitment of Venus-mini-Gs. Thus, NE enters cardiac myocytes, accesses internal βARs, and stimulates PI4P hydrolysis.

## Inhibition of the membrane cation transporter, OCT3, prevents norepinephrine induced PI4P hydrolysis

Data from other laboratories has demonstrated that transport of norepinephrine or epinephrine into cells by organic cation transporter family of proteins (OCT proteins) is required for activation of internal receptors in adult cardiac myocytes (*Wright et al., 2008*) and HELA cells (*Irannejad et al., 2017*). The non-selective cation transporter, OCT3 has been shown to be responsible for this transport, therefore we utilized three structurally distinct inhibitors of OCT3, corticosterone (100 µM, *Figure 5A*), abacavir (10 µM, *Figure 5B*) and lamotrigine (10 µM, *Figure 5C*) (*Dickens et al., 2018*) to determine if OCT3 is required for NE-stimulated intracellular PI4P hydrolysis. Preincubation of NRVMs with any of these inhibitors prevented NE from inducing PI4P hydrolysis, suggesting that OCT3-mediated transport is required for stimulation of PI4P hydrolysisby this catecholamine. In

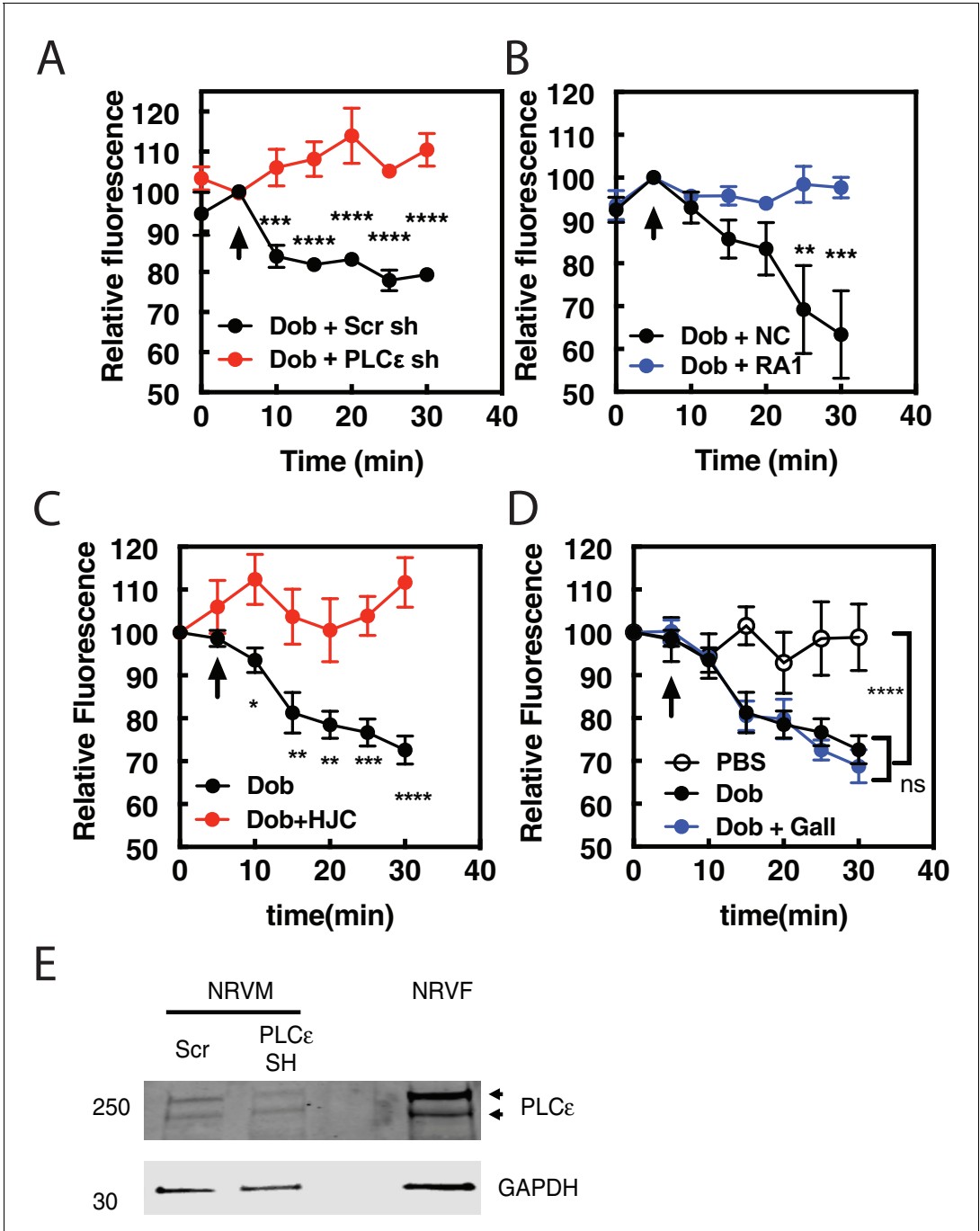

**Figure 2.** Dobutamine-mediated PI4P hydrolysis requires Epac and mAKAPβ bound PLCε. NRVMs were transduced with FAPP-PH-GFP, stimulated with treatments as indicated and Golgi associate fluorescence was monitored with time. (**A**) PLCε knockdown prevents dobutamine-mediated PI4P hydrolysis. NRVMs were transduced for 48 hr with adenovirus expressing shRNA for PLCε or scrambled control shRNA before stimulation with 100 nM dobutamine (at arrow) (N = 3 independent experiments). (**B**) Disruption of PLCε-mAKAPβ interaction prevents PI4P hydrolysis stimulated by dobutamine. NRVMs were transduced with RA1 domain expressing adenovirus or control virus (NC-negative control) as previously described 24 hr before experimentation. Cells were stimulated with 100 nM dobutamine (at arrow) (N = 3 independent experiments). (**C**) Epac is required for dob-mediated PI4P hydrolysis. The Epac inhibitor HJC0726 (1 µM) was added to NRVMs 15 min before imaging and dobutamine (100 nM) was added at the arrow. (N = 3 independent experiments) (**D**) Gβγ is not required for dobutamine-mediated PI4P hydrolysis. The Gβγ inhibitor, Gallein (10 µM) was added 15 min prior to imaging and dobutamine added as indicated by the arrow (N = 3 independent experiments). Data are not significant between Dob and Dob + Gallein. Images for A, B,C and D PI4P hydrolysis were taken from at least 4 cells for each separate preparation of NRVMs. (**E**) Western blot of Adenoviral sh-RNA knockdown of PLCε. The two bands likely represent two splice variants of PLCε.
DOI: https://doi.org/10.7554/eLife.48167.010

*Figure 2 continued on next page*

*Figure 2 continued*

The following source data is available for figure 2:

**Source data 1.** Effects of inhibition of PLCε, Epac and Gβγ on dobutamine stimulated PI4P hydrolysis.
DOI: https://doi.org/10.7554/eLife.48167.011

addition, we sought to determine if receptor internalization is required for NE-mediated PI4P hydrolysis and βAR activation at the Golgi. An inhibitor of dynamin-mediated receptor internalization, Dyngo (40 μM), had no effect on either PI4P hydrolysis (*Figure 5D*) or recruitment of NES-Venus-mini-Gs to the Golgi apparatus (*Figure 5E* and *Figure 5—video 1*) by NE. As a positive control, Dyngo does prevent internalization of FLAG-β1AR at the cell surface (*Figure 5—figure supplement 1A*). Additionally, corticosterone (100 μM) did not alter cAMP production from cell surface βARs (*Figure 5—figure supplement 1B*). Corticosterone did not block dobutamine-mediated PI4P hydrolysis but rather enhanced the rate of onset relative to dobutamine alone (*Figure 5—figure supplement 1C*). The reason for this enhancement is unclear but the differential effects of corticosterone on dobutamine vs NE are consistent with blockade of NE entry into the cell, but not dobutamine. Together, these data indicate that transport by OCT transporters, not receptor internalization, is required for NE-mediated Golgi βAR activation and PI4P hydrolysis.

Additionally, NRVMs were transduced with viruses containing FRB-CFP-Nb80 and FKBP-mApple-GalT, along with FAPP-PH-GFP to determine if Golgi β1AR were required for PI4P hydrolysis to NE. In cells pretreated with vehicle, NE induces PI4P hydrolysis. However, following addition of 1 μM Rapamycin, NE-induced PI4P hydrolysis was significantly inhibited (*Figure 5F*).

To confirm that dobutamine and NE stimulated Golgi PI4P hydrolysis is not unique to neonatal myocytes we tested the ability of dobutamine and NE to stimulate PI4P hydrolysis in acutely isolated murine adult ventricular myocytes. Infection of AVMs for 24 hr with FAPP-PH-GFP labels the Golgi surrounding the nucleus as well as other structures within the myocyte as we have previously reported (*Figure 6A*) (*Malik et al., 2015*). Stimulation of AVMs with dobutamine (*Figure 6B*) or NE (*Figure 6C*) stimulated PI4P depletion in the Golgi. NE-stimulated PI4P hydrolysis was blocked by preincubation with the Oct3 inhibitor abacavir (*Figure 6C*) indicating that NE transport into AVMs is required for stimulation of PI4P hydrolysis, as was seen in NRVMs. These data indicate that internal receptors are required for stimulation of PI4P hydrolysis in AVMs as well as NRVMs.

## Cardiac hypertrophy induced by dobutamine is more effectively inhibited by a cell permeant antagonist

To determine if signaling by internal βARs is important for promotion of cellular hypertrophy, NRVMs were treated with dobutamine (100 nM) for 48 hr and two measures of hypertrophy were assessed, cell area and ANF expression. Dobutamine stimulated an increase cell area (*Figure 7A*), and ANF expression (*Figure 7B*), after 48 hr of stimulation. Co-incubation with the cell permeant antagonist, metoprolol, strongly inhibited these hypertrophic responses. The cell impermeant antagonist, sotalol, on the other hand was significantly less effective at blocking these hypertrophy measures than metoprolol, (*Figure 7A and B*). These data, taken together with PI4P hydrolysis data, suggest that internal βARs are involved in mediating cardiomyocyte hypertrophy.

## NE-stimulated NRVM hypertrophy requires membrane transport by OCT3 and Golgi resident βARs.

NE is an endogenously produced catecholamine involved in mediating hypertrophy and heart failure and stimulation of NRVMs with NE stimulates hypertrophy. To determine the role of intracellular NE in hypertrophy, NRVMs were treated with NE in the presence or absence of corticosterone to block Oct3 cation transporters for 48 hr. Treatment with NE stimulated an increase in cell size (*Figure 8A*) and ANF expression (*Figure 8B*) that was blocked by corticosterone indicating that transport of NE into the myocyte is required to mediate hypertrophy. Treatment with dobutamine also induced hypertrophy but this was not blocked by Oct3 supporting the idea that dobutamine access to internal receptors does not require Oct3 and confirms the specificity of the Oct3 blockade.

To determine if NE and dobutamine driven NRVM hypertrophy requires Golgi βARs NRVMs were transduced with adenovirus containing CFP-Nb80-FRB and FKBP-mApple-GalT for 24 hr followed by

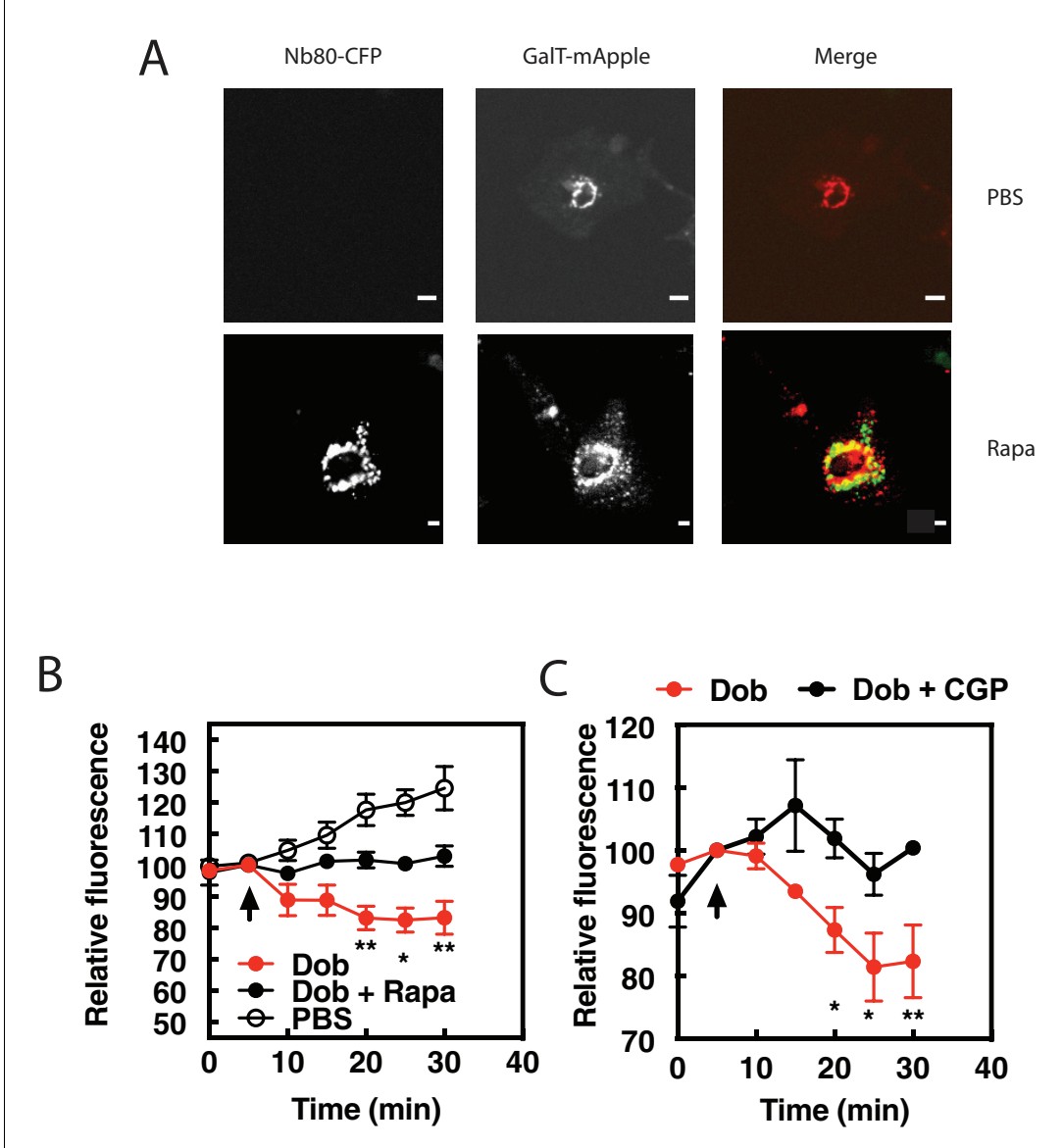

**Figure 3.** β1ARs in the Golgi are required for PI4P hydrolysis stimulated by dobutamine. NRVMs were transduced with FRB-CFP-Nb80 and FKBP-GalT-mApple containing adenovirus, along with adenovirus containing FAPP-PH-GFP for 24 hr. (**A**) Confocal fluorescence images of NRVMs expressing FRB-CFP-Nb80 in the CFP channel and FKBP-GalT-mApple in the red channel before and after addition of rapamycin (see Materials and methods for details) Pearson's correlation coefficient = 0.32 ± 0.03. Scale bars are 10 μm. (**B**) NRVMs were incubated with either rapamycin (1 μM) or DMSO control for 15 min prior to addition of dobutamine (100 nM) added at the arrow. Golgi associated FAPP-PH-GFP fluorescence was monitored by time lapse fluorescence video microscopy as in *Figure 1A*. (**C**) Cells were pretreated treated with the cell permeant β1AR selective antagonist, CGP-20712 (100 nM), or vehicle, followed by dobutamine addition and assessed as in B. Images were taken from at least n = 9 cells each from at least four separate preparations of NRVMs. Data were analyzed as means from N = 4 experiments. Agonists were added as indicated by the arrow.
DOI: https://doi.org/10.7554/eLife.48167.012

The following source data is available for figure 3:

**Source data 1.** Effects of Golgi targeted NB80 and β1AR-specific antagonism on dobutamine stimulated PI4P hydrolysis.
DOI: https://doi.org/10.7554/eLife.48167.013

treatment for 48 hr with NE or dobutamine with and without co-treatment with rapamycin. In the absence of rapamycin, NE and dobutamine stimulated increases in cell area and ANF expression (*Figure 8C and D*). Co-treatment with rapamycin to translocate Nb80 to the Golgi blocked NE and

dobutamine stimulated hypertrophy, demonstrating that Golgi βARs are required for stimulation of hypertrophy by these agonists.

## Discussion

We previously found that raising cAMP levels with forskolin in NRVMs stimulated PI4P hydrolysis, but Iso did not unless a specific PDE, PDE3 was blocked. This led to the question of how the Epac/mAKAPβ/PLCε complex could be activated during physiological cAMP regulation by Gs coupled receptors. We considered two potential mechanisms, one where chronic agonist exposure alters the organization of PDEs in the cardiac myocyte such that Epac could be accessed by global cAMP changes, or that intracellular βARs generate cAMP with privileged access to the Epac/mAKAPβ/PLCε complex. Here we demonstrate that stimulation of Golgi resident β1ARs induces Epac-PLCε-dependent PI4P hydrolysis when stimulated by either a membrane permeant drug, dobutamine, or the physiological neurotransmitter, norepinephrine. We also show that stimulation of these internal receptors is required for both dobutamine and NE stimulated cardiomyocyte hypertrophy.

### Compartmentalized GPCR signaling

It is well established that GPCRs can be found on intracellular compartments (*Jong et al., 2018*). A number of investigators have characterized GPCRs on the nuclear envelope in neurons and cardiac myocytes, and signaling mechanisms have been proposed (*Kumar et al., 2008*; *Boivin et al., 2006*; *Merlen et al., 2013*; *Dahl et al., 2018*). GPCRs and G proteins have been found on other intracellular compartments including the Golgi apparatus, but signaling roles for receptors at the Golgi are not well defined. This is in part because GPCRs are trafficked through the Golgi apparatus on their way to the cell surface and it has not been clear that GPCRs in the Golgi have a signaling function. The recent work by *Irannejad et al. (2017)* definitively demonstrated that Golgi localized β1ARs and Gs can be driven to an active conformation by exogenous ligands. Recent work from the Calebiro laboratory demonstrated that the thyroid stimulating hormone (TSH) receptor activates Gs in the Golgi in mouse thyroid cells through a mechanism that relies on endosomal retrograde trafficking of internalized receptors (*Godbole et al., 2017*). This is in contrast to what is shown here in cardiac myocytes, and as reported by Irannejad et al. in Hela cells, where activation of Golgi β1ARs does not involve receptor internalization but rather relies on Golgi resident receptors and transport of ligands across the cell membrane via Oct3.

The work presented here elucidates a possible physiological function for Golgi resident βARs through a signal transduction pathway we previously demonstrated to be involved in regulation of cardiac hypertrophy (*Figure 9*) (*Zhang et al., 2011*; *Zhang et al., 2013*; *Malik et al., 2015*; *Nash et al., 2018*). This pathway involves a nuclear envelope scaffolded complex of Epac/PLCε and mAKAPβ. cAMP dependent activation of Epac acts an exchange factor for the small GTPase Rap which directly stimulates PLCε enzymatic activity toward Golgi PI4P, resulting in local production of DAG and subsequent nuclear PKD activation. A conundrum arising from previous work was that direct activation of Epac with a cAMP analog cpTOME, or treatment with forskolin strongly stimulates PLCε dependent Golgi PI4P hydrolysis, however; stimulation with Iso had no effect on PI4P hydrolysis despite strongly stimulating global cAMP production. Demonstration that a Golgi resident β1AR can locally activate this system solves this conundrum. Missing from this analysis is demonstration of the presence of adenylyl cyclase (AC) at the Golgi. Due to the inadequacy of AC antibodies and the likely low levels of this enzyme, localization of AC by immunocytochemical analysis is not possible. Nevertheless, binding of AC5 to mAKAPβ in cardiac myocytes has been previously demonstrated (*Kapiloff et al., 2009*) indicating that AC is likely locally available to be activated by Golgi β1ARs and Gs.

The purpose of internal GPCR signaling is likely to generate signals that are local and distinct from receptors at the PM or other intracellular locations. In this way specific processes can be controlled by second messengers that have the potential to activate a vast number of global responses if the signals were not restricted. In the case of the cardiac myocyte, during acute sympathetic stimulation, norepinephrine may only access cell surface βARs needed to mediate increases in cardiac contraction required to meet the demands of a sympathetic response. Because signaling by Golgi βARs requires uptake of ligand prior to activation, these processes are slower and may not be accessed during acute sympathetic responses, but rather are only activated during more chronic exposure

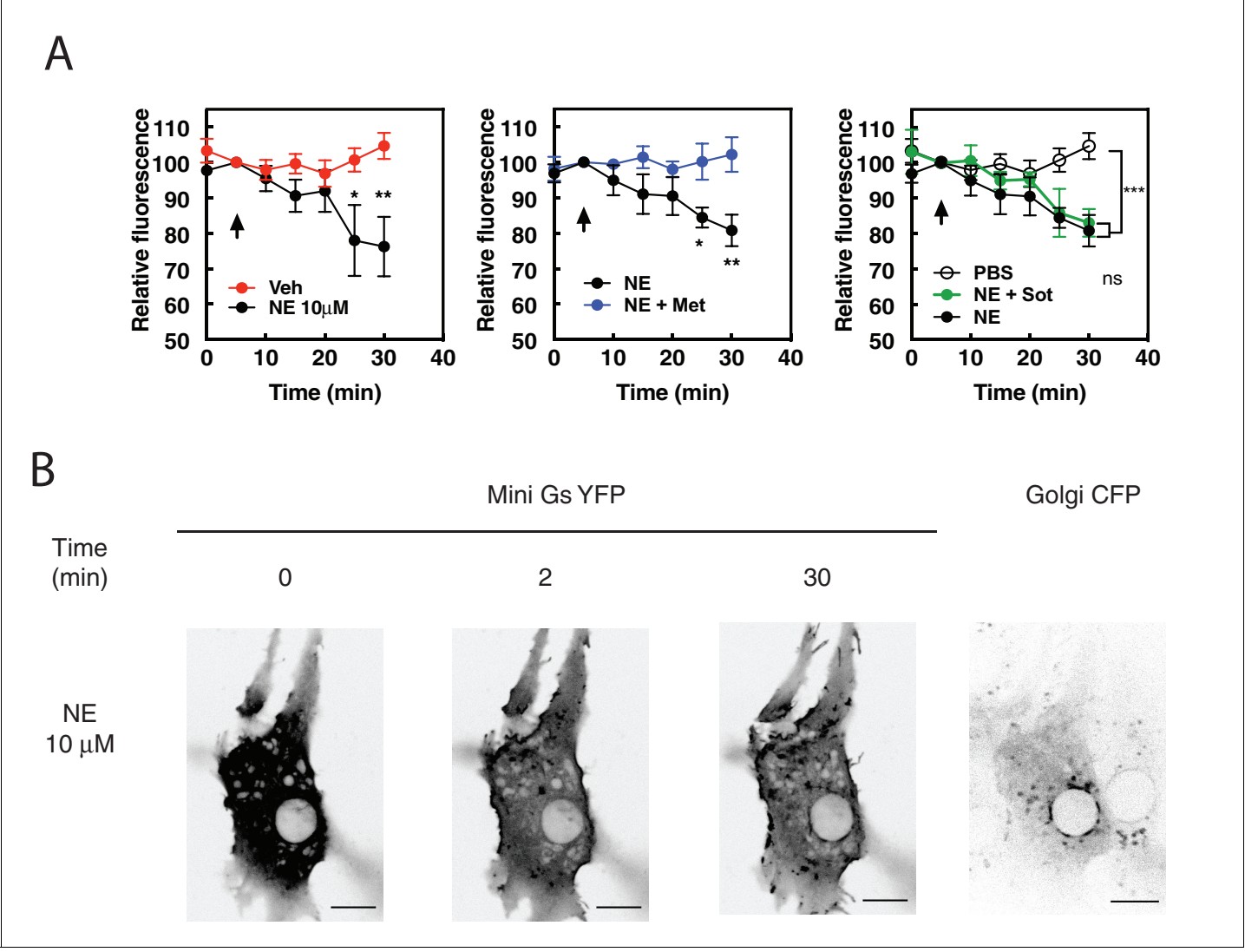

**Figure 4.** The physiological neurotransmitter, norepinephrine, can induce PI4P hydrolysis through internal receptors and can activate βARs at the Golgi. (A) NRVMs were transduced with FAPP-PH-GFP and stimulated with norepinephrine (10 μM, left) in the presence of metoprolol (100 μM, center) or sotalol (5 mM, right) and analyzed as in *Figure 1A*. Data are not significant between norepinephrine and norepinephrine + sotalol. (B) NRVMs were transfected with β1AR and NES-Venus-mini-Gs, followed by viral transduction with CFP-Giantin. Representative images of norepinephrine-mediated NES-. Venus-mini-Gs recruitment (10 μM, left columns), CFP-Giantin Golgi marker (B, right). The Pearson's Coefficient for colocalization of Venus-mini-Gs and CFP-Giantin is R = 0.455 ± 0.06. Scale bars are 10 μm. All experiments were performed in humidified environmental chamber at 37°C. Images for PI4P hydrolysis were collected as in *Figure 1A*, and were from at least n = 7 cells each from three separate preparations of NRVMs. Data were analyzed as means from N = 3 experiments. Agonists were added where indicated by the arrow.

DOI: https://doi.org/10.7554/eLife.48167.014

The following video and source data are available for figure 4:

**Source data 1.** NE stimulates PI4P hydrolysis.
DOI: https://doi.org/10.7554/eLife.48167.015
**Figure 4—video 1.** Norepinephrine activates β1ARs in the Golgi apparatus.
DOI: https://doi.org/10.7554/eLife.48167.016
**Figure 4—video 2.** Norepinephrine activates β1ARs in the Golgi apparatus.
DOI: https://doi.org/10.7554/eLife.48167.017

such as is observed in cardiovascular stress. Separation of the Epac/mAKAPβ/PLCε pathway at the nuclear envelope/Golgi interface from global cAMP dependent PKA activation possibly prevents inappropriate activation of PLCε dependent hypertrophic responses to acute βAR activation.

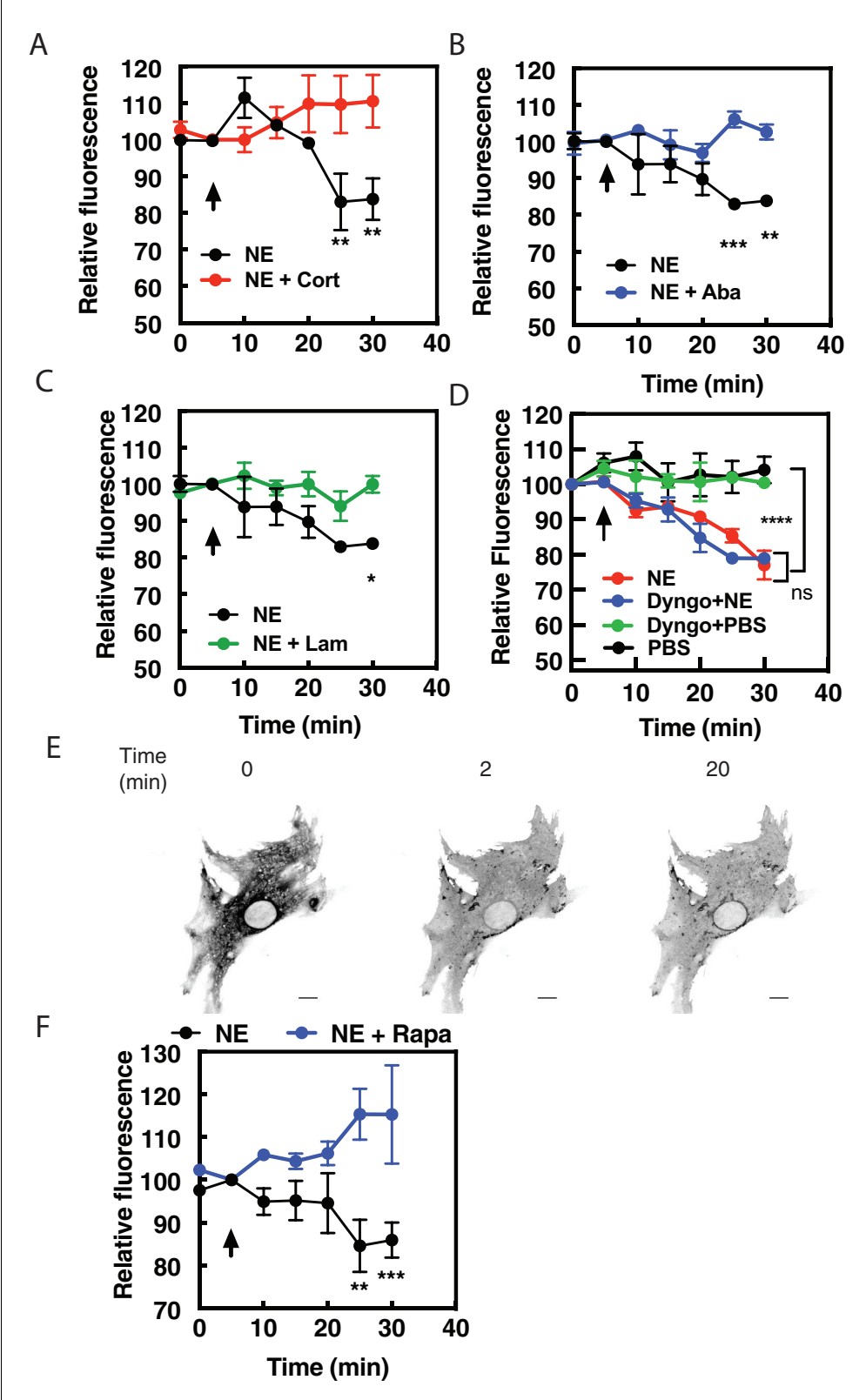

**Figure 5.** Norepinephrine requires OCT transporters but not receptor internalization to stimulated PI4P hydrolysis. NRVMs were transduced with FAPP-PH-GFP and stimulated with norepinephrine (10 µM) in the presence of corticosterone (100 µM, (**A**), abacavir (10 µM, (**B**), lamotrigine (10 µM, (**C**) or Dyngo (40 µM, (**D**). Data are not significant between norepinephrine and norepinephrine + Dyngo. Images were collected as in *Figure 1A* for PI4P hydrolysis (N = 3) were from at least 4 cells each from separate preparations of NRVMs. Agonists were added where indicated by the arrow. (**E**) Dyngo

*Figure 5 continued on next page*

*Figure 5 continued*

has no effect on NES-Venus-mini-Gs Golgi recruitment by norepinephrine. Representative image of norepinephrine-mediated NES-Venus-mini-Gs recruitment in the presence of Dyngo (40 μM). (N = 3) Scale bars are 10 μm. (**F**) NRVMs were transduced with adenovirus containing FRB-CFP-Nb80 and FKBP-GalT-mApple along with adenovirus containing FAPP-PH-GFP for 24 hr prior to experimentation. NRVMs were incubated with either rapamycin (1 μM) or DMSO control for 15 min prior to addition of NE (10 μM) added at the arrow. Images were collected as in *Figure 1A* for PI4P hydrolysis and were from at least n = 10 cells each from three separate preparations of NRVMs. Data were analyzed as means from N = 3 experiments.

DOI: https://doi.org/10.7554/eLife.48167.018

The following video, source data, and figure supplement are available for figure 5:

**Source data 1.** NE requires membrane transport by OCT3 and Golgi resident βARs to stimulate PI4P hydrolysis.
DOI: https://doi.org/10.7554/eLife.48167.020
**Figure supplement 1.** Receptor internalization does not contribute to Golgi β1AR localization.
DOI: https://doi.org/10.7554/eLife.48167.019
**Figure 5—video 1.** Inhibition of receptor internalization does not alter activation of β1ARs in the Golgi apparatus.
DOI: https://doi.org/10.7554/eLife.48167.021

## Requirement for internal β1ARs for cardiac hypertrophy

Significantly, stimulation of hypertrophy by the natural sympathetic ligand NE required activity of Golgi localized βARs. As has been discussed, sustained elevated sympathetic drive is one of the major factors responsible for development of cardiac hypertrophy due to chronic hypertension. β-blocker therapy efficacy is thought to result from preventing chronic stimulation of cell surface βARs by elevated catecholamines and subsequent desensitization (*Bristow, 2000*). Our results suggest that blockade of internal receptors may be required to prevent catecholamine-induced hypertrophy and demonstrates that activation of cell surface GPCRs is not sufficient for natural catecholamines to induce hypertrophy in NRVMs. The importance of blockade of Golgi GPCRs is supported by the fact that the cell permeant βAR antagonist metoprolol is significantly more effective than the cell impermeant antagonist sotalol in prevention of dobutamine stimulated cardiomyocyte hypertrophy. Interestingly, clinically effective β-blockers tend to be relatively hydrophobic. Metoprolol and carvedilol are first line β blockers that are used to treat heart failure and have log P values 1.8 and 4.2 respectively, and are thus relatively cell permeant. Sotalol, which has a log P of −0.85 is relatively cell impermeant, and is not clinically used for heart failure treatment. We speculate that hydrophobicity might be an important factor for β-blocker development allowing these inhibitors to access internal pools of receptor.

A seeming contradiction is the observation that the cell impermeant agonist isoproterenol induces hypertrophy in vitro and when chronically administered in vivo (*Zhang et al., 2011*; *Iaccarino et al., 1998*). Iso administered in vivo is not at physiological concentrations and has multiple target tissues beyond the cardiac myocyte, such as the vasculature, that could contribute to hypertrophy. For NRVMs in vitro, while Iso can induce hypertrophy by a mechanism that probably does not rely on internal receptors, the effects of the natural ligands epinephrine and norepinephrine are more relevant to the in vivo situation and do rely on Golgi localized βARs. The synthetic agonist Iso seems to rely on an alternative PM mediated pathway that is not engaged by the physiological agonist NE. It will be critical to determine if in the in vivo pathology of heart failure that internal β receptors play a critical role using animal models such as transaortic constriction (TAC) that mimic hypertensive stress.

## Materials and methods

### Key resources table

| Reagent type (species) or resource | Designation | Source or reference | Identifiers | Additional information |
|---|---|---|---|---|
| Strain, strain background (Rat) | NRVM | | | Freshly isolated myocytes from neonatal rats |

*Continued on next page*

*Continued*

| Reagent type (species) or resource | Designation | Source or reference | Identifiers | Additional information |
|---|---|---|---|---|
| Recombinant DNA reagent | NES-Venus-mini-Gs | Nevin Lambert, Augusta University, GA. *Wan et al. (2018)* | | |
| Recombinant DNA reagent (human) | FLAG-β1 adrenergic receptor | Addgene | RRID: Addgene_14698 | |
| Recombinant DNA reagent | CFP Giantin | *Irannejad et al. (2017)* | | Golgi Marker Adenovirus expressing CFP Giantin MOI 50 |
| Transfected construct (Rat) | PLCε-RA1 | *Zhang et al. (2013)* | | Adenovirus expressing the RA1 domain from *PLCE1* MOI 50 |
| Transfected construct (Rat) | *PLCE1*-shRNA | *Zhang et al. (2013)* | | Adenovirus expressing *PLCE1* shRNA MOI 50 |
| Transfected construct | FAPP-PH-GFP | *Zhang et al. (2013)* | | Adenovirus expressing FAPP-PH-GFP for PI4P detection MOI 50 |
| Transfected construct | CFP-Nb80-FRB | This paper and *Irannejad et al. (2017)* | | In this paper adenovirus was created for expressing CFP-Nb80-FRB previously created in *Irannejad et al. (2017)* MOI 50 |
| Transfected construct | FKBP-mApple-GalT | This paper and *Irannejad et al. (2017)* | | In this paper adenovirus was created for expressing FKBP-mApple-GalT previously created in *Irannejad et al. (2017)* MOI 50 |
| Antibody | Anti-β1 adrenergic receptor (rabbit polyclonal) | Abcam | #ab3442 RRID: AB_10890808 | 1:100 dilution |
| Antibody | Anti-TGN38 (sheep anti Rat polyclonal) | Biorad | #AHP499G RRID:AB_2203272 | 1:1000 dilution |
| Antibody | M2-FLAG-Cy3 (mouse monoclonal) | Sigma | A9594 RRID:AB_439700 | 5 ug/mL |
| Chemical compound, drug | Butanedione -monoxime (BDM) | Sigma | 112135 | Myosin blocker |
| Chemical compound, drug | Isoproterenol (ISO) | Sigma | 1351005 | βAR agonist |
| Chemical compound, drug | Dyngo | Abcam | Ab120689 | Dynamin inhibitor |
| Chemical compound, drug | Sotalol | Sigma | S0278 | βAR antagonist |
| Chemical compound, drug | HJC0726 | Xiaodong Cheng, UT Houston Health Science Center. *Zhu et al. (2015)* | | Epac inhibitor |
| Chemical compound, drug | Brefeldin A | Biolegend | 420601 | Golgi disruptor |
| Chemical compound, drug | Gallein | Sigma | 371708 | G protein βγ subunit inhibitor |

*Continued on next page*

*Continued*

| Reagent type (species) or resource | Designation | Source or reference | Identifiers | Additional information |
|---|---|---|---|---|
| Chemical compound, drug | Corticosterone | Tocris | 3685 | Oct3 inhibitor |
| Chemical compound, drug | Metoprolol | Sigma | M5391 | βAR antagonist |
| Chemical compound, drug | Dobutamine | Tocris | 0515 | βAR agonist |
| Chemical compound, drug | Lamotrigine | Tocris | 2289 | OCT3 blocker |
| Chemical compound, drug | Abacavir | Tocris | 4148 | OCT3 blocker |
| Chemical compound, drug | Norepinephrine (NE) | Sigma | A0937 | AR agonist |
| Chemical compound, drug | Rapamycin (Rapa) | Tocris | 1292 | |
| Peptide, recombinant protein | Collagenase Type II | Worthington | CLS-2 | |
| Commercial assay or kit | cAMP Elisa | ENZO | ADI-900–066 | |

## Isolation of neonatal cardiac myocytes and adenoviral transduction

Briefly, hearts were excised from 2 to 4 day old Sprague-Dawley rats, ventricles separated and minced thoroughly before digestion with Collagen type II (Worthington) in Hanks buffered saline solution (HBSS) without $Ca^{2+}$ or $Mg^{2+}$. Following digestion, cells were collected by centrifugation into Dulbecco Modified Eagle Medium (DMEM) supplemented with 10% fetal bovine serum (FBS), 100 U/mL penicillin, 100 µg/mL streptomycin, 2 mM glutamine and 2 µg/mL vitamin B12. Contaminating cells were removed by preplating cells onto tissue culture plastic for a minimum of 1 hr at 37°C. NRVMs were then plated onto either glass-bottom tissue culture plates or 12 well plates coated with 0.2% gelatin and cultured in DMEM (composition as above, with additional 10 µM cytosine arabinoside). 48 hr later, cells were transferred into media supplemented with 1% FBS. For transduction with adenovirus, 50 MOI of indicated adenovirus was added overnight upon transfer to 1% FBS containing media. For fluorescent adenovirus constructs, expression was confirmed the next day by epifluorescence microscopy.

## Isolation of adult ventricular myocytes

Adult myocytes were isolated from 2 to 4 month-old wild type C57BL/6 mice as previously described (*Auerbach et al., 2013*). Briefly, mice were anesthetized with ketamine (100 mg/kg body weight) and xylazine (5 mg/kg body weight) i.p.. Hearts were cannulated and perfused with perfusion buffer via the aorta. Subsequently, the hearts were perfused with digestion buffer consisting of: collagenase type-II (773.48 U/mL), trypsin (0.14 mg/mL), and calcium chloride (12.5 µM) in the perfusion buffer (pH 7.46). The atria were removed and the ventricles were minced in stop buffer containing 10% FBS and 12.5 µM calcium chloride in perfusion buffer. After calcium was added back to a final concentration of 1 mM, cells were plated onto laminin-coated 20 mm glass bottom dishes in minimum essential medium supplemented with 0.35 g/L sodium bicarbonate, 2.5% FBS, and 10 mM 2,3 butanedione monoxime (BDM).

## Transduction of AVMs with adenovirus

After AVMs were adhered to the laminin-coated 20 mm dishes for 1 hr, plating media was removed and cells were transduced with adenoviruses expressing FAPP-PH-GFP (100 MOI) in BDM-free media for 2–3 hr. The virus was then removed and BDM was added to the culture media. 18–24 hr later, cells were imaged by confocal microscopy with a 40X oil-immersion lens for measurement of FAPP-PH-GFP fluorescence. EGFP was excited at 488 nm and images were acquired with 25 ms exposures at 2.5 min intervals.

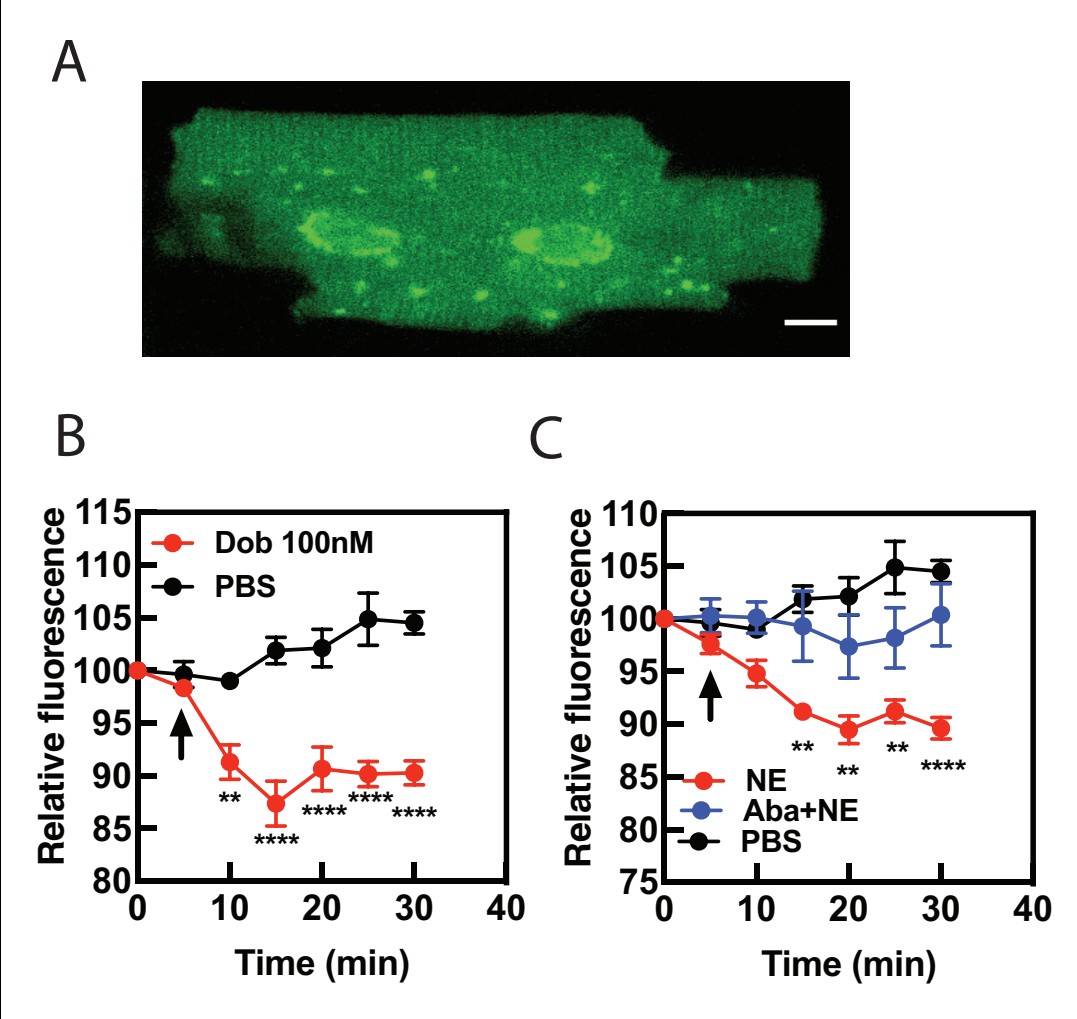

**Figure 6.** Dobutamine and NE stimulate PI4P hydrolysis in Adult Ventricular Myocytes (AVMs). Freshly isolated mouse AVMs were infected with FAPP-PH-GFP (100 MOI) for 24 hr. (**A**) representative image of an AVM expressing FAPP-PH-GFP showing strong labeling surrounding the nucleus. (**B**) FAPP-PH-GFP expressing AVMs were treated with either dob or PBS control as indicated and were imaged as for NRVMs as in *Figure 1A* for PI4P hydrolysis. Fluorescence intensity in the region surrounding the nucleus was quantitated at 5 min intervals. **C**) AVMs were treated with PBS control, NE (10 μM) or NE plus abacavir and analyzed as in B. Data for B and C are from at least n = 3 cells each from three separate preparations of AVMs. Data were analyzed as means from N = 3 experiments.
DOI: https://doi.org/10.7554/eLife.48167.022
The following source data is available for figure 6:

**Source data 1.** NE-Stimulated PI4P hydrolysis in adult cardiac myocytes requires OCT3.
DOI: https://doi.org/10.7554/eLife.48167.023

## NES-Venus-mini-Gs imaging

NRVMs were plated into gelatin-coated 20 mm glass bottom cell culture dishes. Cells were transfected the following day with plasmids (500–800 ng of β1-ARs and 250–400 ng of NES-Venus-mini-Gs per dish) using lipofectamine 3000. Media was changed to 1% FBS the next day and transduced with adenovirus-expressing CFP- Giantin overnight. Cells were imaged in confocal mode with a Leica DMi8 equipped with a Crest-optics X-light V2 confocal unit and a 100 × 1.4 NA oil-immersion lens. Venus was excited at 515 with an X-Cite Xled1 light source, and emission monitored imaged on a backlit CMOS Photometrics Prime 95B camera.

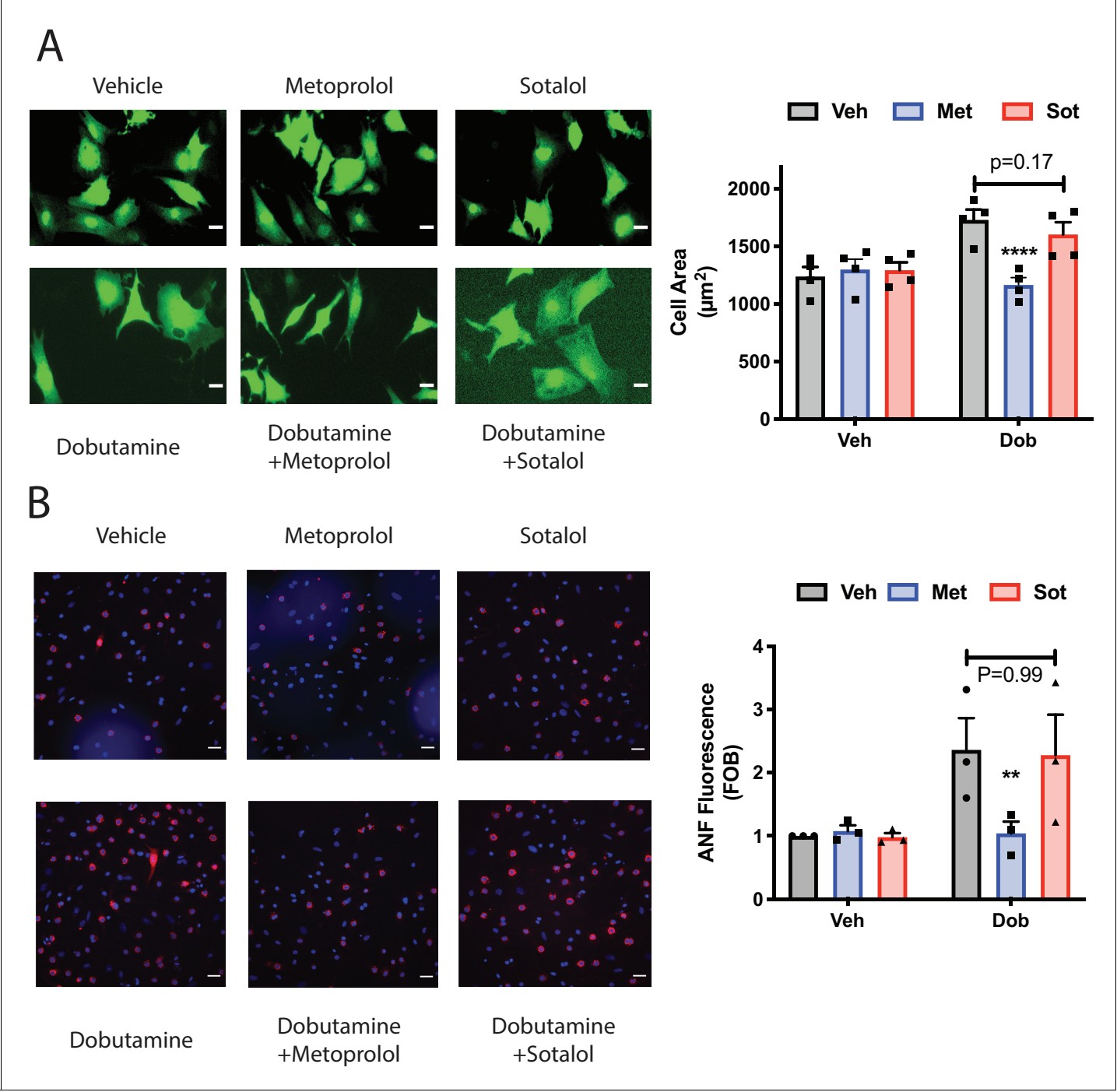

**Figure 7.** Dobutamine induced cardiomyocyte hypertrophy requires intracellular βARs. (**A**) Dobutamine induces internal-receptor dependent increases in cell area. NRVMs were transduced with YFP virus prior to stimulation for 48 hr with dobutamine in the presence of the indicated antagonists or vehicle control. Following fixation, cell area was measured using image J. Representative images are on the left with mean data (fold over basal, FOB) on the right (N = 4 separate preparations of NRVMs), analyzed with a paired two way ANOVA. ****p<0.0001 vs. Dob and Dob+sotalol. Scale bars are 10 μm. **B**) Dobutamine induces an increase in ANF expression via an internal receptor-dependent mechanism. NRVMs were stimulated with dobutamine in the presence or absence of the indicated antagonists or vehicle control for 48 hr before fixation and staining for ANF. Fluorescence of ANF rings was then captured by confocal microscopy, followed by fluorescence intensity analysis with Image J. Representative images are on the left with mean data (fold over basal, FOB) on the right (N = 3 separate preparations of NRVMs), analyzed with a paired two way ANOVA. **p=0.007 vs. Dob and Dob+sotalol. The total number of cells analyzed was greater than 200 cells for each.

DOI: https://doi.org/10.7554/eLife.48167.024

*Figure 7 continued on next page*

*Figure 7 continued*

The following source data is available for figure 7:

**Source data 1.** Dobutamine-stimulated cardiomyocyte hypertrophy is blocked by a cell permeable βAR antagonist.

DOI: https://doi.org/10.7554/eLife.48167.025

## Measurement of PI4P hydrolysis

Measurements of PI4P hydrolysis were made as previously described (*Zhang et al., 2013*; *Malik et al., 2015*; *Nash et al., 2018*). After preparation and culture of myocytes, cells were transduced with adenovirus (50 MOI) expressing GFP-FAPP-PH overnight. The following day, expression was confirmed by epifluorescence microscopy. Time lapse video fluorescence Imaging of GFP-FAPP-PH fluorescence was performed at room temperature or 37℃, where indicated, on a LEICA DMi8 with a 20x air lens in confocal mode. EGFP was excited at 488 nm with an X-Cite Xled1 light source and emission monitored imaged on a backlit CMOS Photometrics Prime 95B camera. Images were acquired with 50 ms exposure times at 1 min intervals to minimize photobleaching. Analysis of fluorescence intensity changes from the videos was performed using NIH Image J unless otherwise stated. Analysis was performed by subtracting background fluorescence intensity from a region of interest intensity at all time points measured. Data is presented as percentage of fluorescence remaining after agonist stimulation when compared to cells prior to stimulation.

## Measurement of myocyte cell area

NRVMs were plated into gelatin-coated 12 well plates or glass-bottom 96 well plates and allowed to grow overnight. The following day, cells were infected overnight with adenovirus expressing YFP. Subsequently, NRVMs were stimulated with dobutamine (100 nM) or norepinephrine (10 µM) for 48 hr, in the presence of antagonists or transduction with FRB-CFP-Nb80 and FKBP-mApple-GalT, as indicated. Following stimulation, cells were fixed in 4% (w/v) paraformaldehyde. Fluorescent images were taken at 10 x magnification and cell area measured using NIH Image J or Cell Profiler software from over 500 cells from at least three separate experiments.

## Immunocytochemistry for ANF induction

NRVMs were plated into gelatin-coated eight chamber glass slides or glass-bottom 96 well plates and allowed to grow overnight. The following day, cells were serum starved for 24 hr. Subsequently, NRVMs were stimulated with dobutamine (100 nM) or norepinephrine (10 µM) for 48 hr, in the presence of antagonists or transduction with FRB-CFP-Nb80 and FKBP-mApple-GalT, as indicated. Cells were washed with PBS and fixed with 4% PFA for 15 min and then incubated with 10% normal goat serum in phosphate buffered saline containing 0.1% Triton X100 (PBS-T) for 1 hr at room temperature. Primary antibody was incubated at a dilution of 1:1000 in 2% goat serum in PBS-T overnight at 4C. After three washes with PBS-T, cells were incubated with secondary antibody (Goat anti-rabbit Alexa Fluor 568) at a dilution of 1:1000 in PBS-T for 1.5 hr at room temperature. After three washes with PBS-T, DAPI was added at a dilution of 1:500 in PBS and incubated for 30 min. Fluorescence images were captured at 10 x magnification and fluorescence intensity corresponding to ANF staining surrounding the nucleus was quantified using either NIH Image J or Cell Profiler software.

## Inhibition of Golgi βARs by Nb80

NRVMs were plated onto 20 mm glass bottom coverslips and allowed to adhere for 24 hr. Following adhesion, cells were transduced with adenoviruses containing FKBP-mApple-GalT construct and FRB-CFP-Nb80. The following day, NRVMs were transduced with adenovirus (50 MOI) expressing GFP-FAPP-PH overnight. 15 min prior to experimentation, cells were treated with either 1 µM rapamycin or DMSO, where indicated, to induce translocation of Nb80 to the Golgi membrane. Imaging of GFP-FAPP-PH fluorescence was performed at 37℃ on a LEICA DMi8 in confocal mode with a 20 x air lens. Analysis was performed as for PI4P hydrolysis as previously indicated.

## Saponin permeabilization and visualization of Nb80 translocation

NRVMs were plated onto 20 mm glass bottom coverslips and allowed to adhere for 24 hr. Following adhesion, cells were transduced with adenoviruses containing FKBP-mApple-GalT construct and

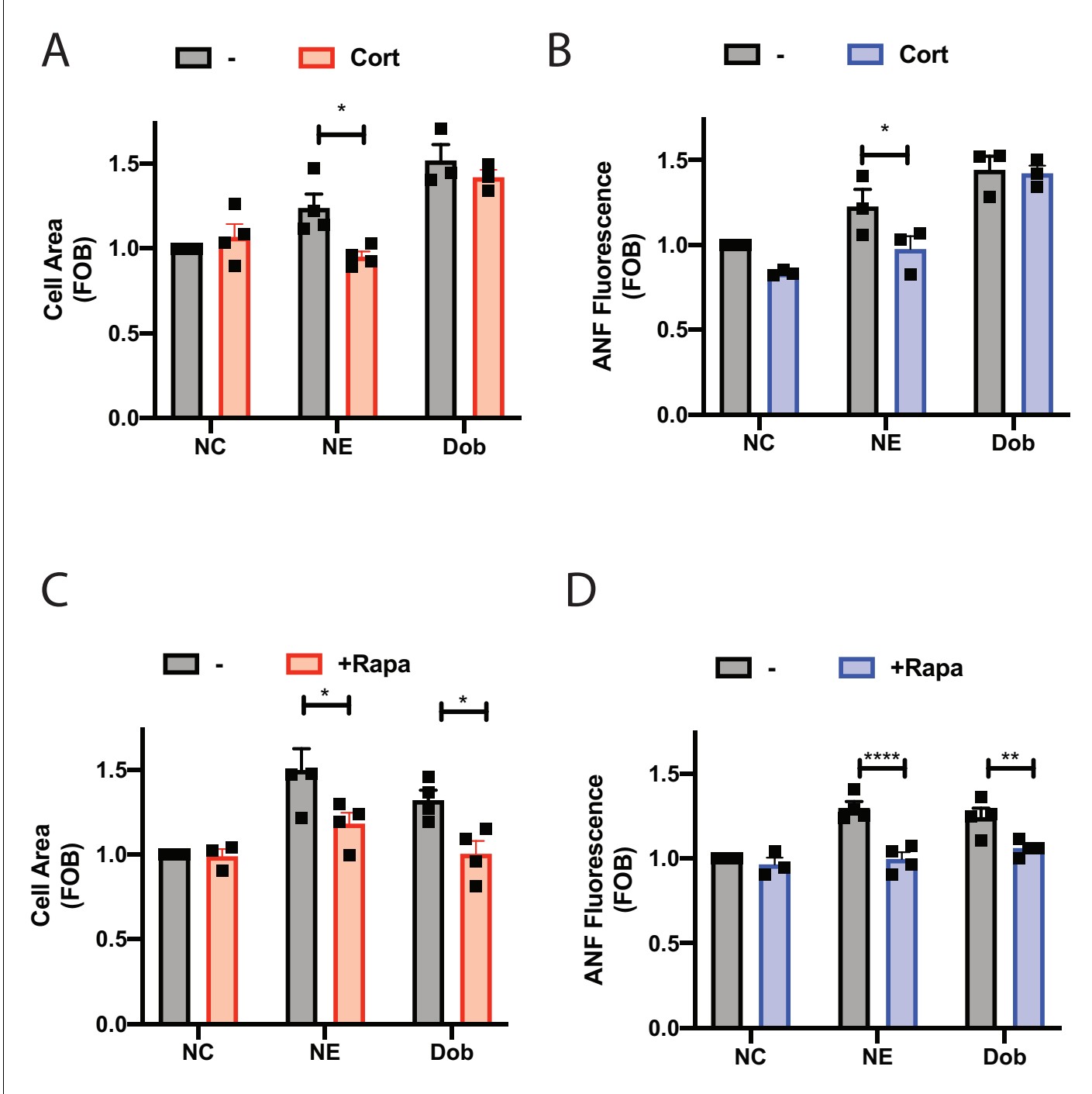

**Figure 8.** Dobutamine and norepinephrine induced cardiomyocyte hypertrophy requires Golgi-localized βARs. Norepinephrine induced hypertrophy also requires agonist internalization. Norepinephrine induces internal-receptor dependent increases in cell area (A) and ANF expression (B). NRVMs were stimulated for 48 hr with norepinephrine (10 μM) or dobutamine (100 nM) in the presence of Corticosterone (100 μM) or vehicle control. Following fixation, cells were stained for ANF and using CellTracker Deep Red. Images were captured using a Thermo Fisher Cell Insight and analyzed by Cell Profiler. Both NE and Dobutamine significantly increase cell size (NE, p<0.05 and Dob p<0.001) and ANF expression (NE, p<0.01 and Dob p<0.001) in control conditions. Dobutamine and Norepinephrine require Golgi localized βAR to induce increase in cell area (C) and ANF expression (D). NRVMs were transduced with FRB-CFP-Nb80 and FKBP-mApple-GalT for 24 hr before stimulation. 15 min before agonist addition, Rapamycin or (1 μM) or vehicle control was added. Cells were then stimulated with either dobutamine (100 nM) or norepinephrine (10 μM) for 48 hr. Following fixation, cells were stained for ANF and using CellTracker Deep Red. Images were captured using a Thermo Fisher Cell Insight and analyzed by Cell Profiler. Both NE

*Figure 8 continued on next page*

*Figure 8 continued*

and dobutamine significantly increase cell size (NE, p<0.001 and Dob p<0.01) and ANF expression (NE, p<0.001 and Dob p<0.001) in control conditions. All data is from at least 1200 cells from three separate preparations of NRVMs. Data were analyzed as means from N = 3-4 experiments.

DOI: https://doi.org/10.7554/eLife.48167.026

The following source data is available for figure 8:

**Source data 1.** NE-stimulated hypertrophy requires OCT3 and Golgi resident βARs.

DOI: https://doi.org/10.7554/eLife.48167.027

FRB-CFP-Nb80. After 48 hr of transduction, cells were treated with either 1 μM rapamycin or PBS control for 15 min. Following treatment, cells were permeabilized with PEM/saponin buffer (80 mM K-PIPES, pH 6.8, 5 mM EGTA, 1 mM MgCl$_2$, 0.05% saponin) for 5 min on ice. Subsequently, cells were fixed with 4% formaldehyde and imaged on a LEICA DMi8 microscope in confocal mode with a 20 x air lens.

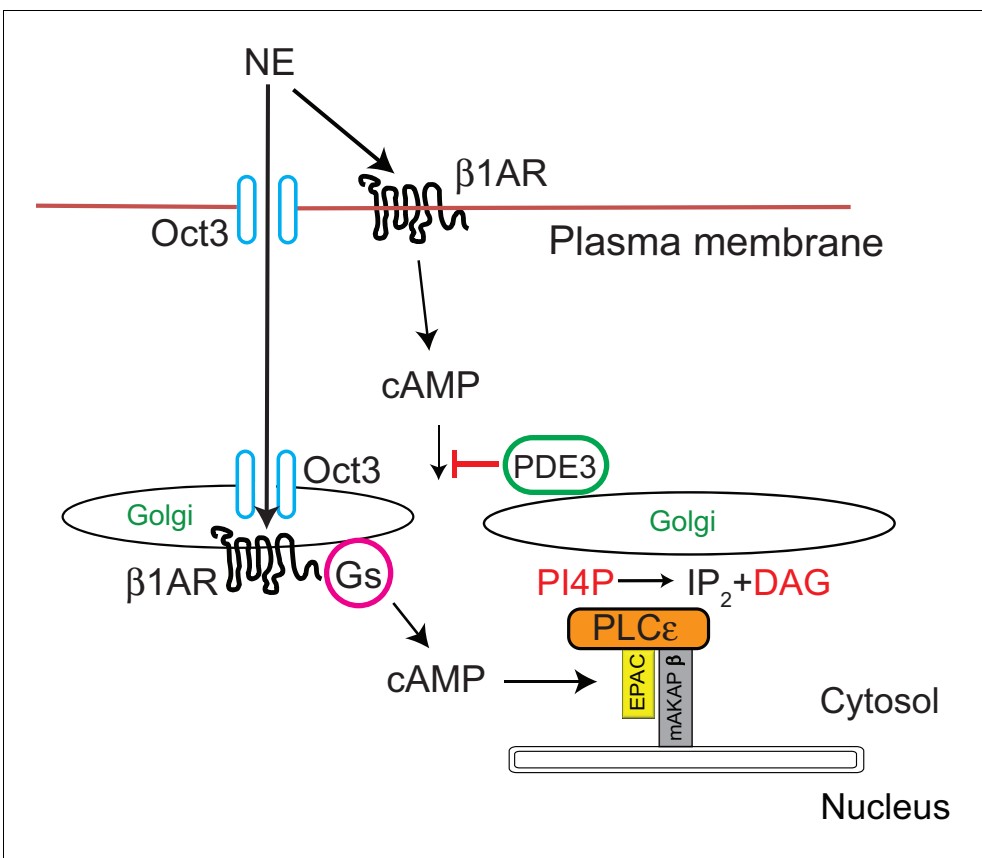

**Figure 9.** Signal transduction by cell surface and Golgi β1ARs. β1ARs are located on both the plasma membrane and the Golgi apparatus in cardiac myocytes. Stimulation of cell surface β1ARs leads to production of cytosolic cAMP but this cAMP cannot access the Epac/PLCε/mAKAPβ due to PDE3 dependent hydrolysis of cAMP. To access Golgi β1AR, NE crosses the plasma membrane via the Oct3 transporter. We speculate that the Oct3 transporter is also present in the Golgi allowing access to the Golgi lumen and there is evidence for the presence of Oct3 on intracellular membranes as discussed in the text. Once activated in the Golgi, β1AR stimulates Gs and subsequent production of cAMP locally. Missing from this diagram is adenylyl cyclase which we presume is in the Golgi apparatus. There is evidence for adenylyl cyclase binding to mAKAPβ in cardiac myocytes. Local cAMP has privileged access to the Epac/PLCε/mAKAPβ complex which leads to PLCε-dependent production of local DAG from PI4P.

DOI: https://doi.org/10.7554/eLife.48167.028

## Internalization of $\beta_1$-adrenergic receptor

NRVMs were plated onto 20 mm glass bottom coverslips and allowed to adhere for 24 hr. Following adhesion, cells were transfected with 500 ng of FLAG-$\beta_1$-AR and allowed to express for 48 hr. The day of the experiment, cells were incubated with 5 µg/ml of M2-FLAG-Cy3 antibody for 1 hr at 37°C. Subsequently, cells were stimulated with agonists as indicated for 30 mins, in the presence or absence of 40 µM Dyngo. Cells were then washed twice with ice-cold HBSS before fixation in 4% formaldehyde. Cells were imaged on a LEICA DMi8 microscope in confocal mode with 63x oil lens.

## cAMP ELISA

NRVMs were plated onto 96 well plates and grown for a minimum of two days before stimulation. Cells were washed twice with HBSS ($+Ca^{2+}/Mg^{2+}$) and incubated with IBMX (300 µM) for 10 mins before addition of inhibitors for a further 10 mins, as indicated. Agonist stimulations were performed in HBSS buffer for 10 min before termination of assay with 0.1 M HCl. cAMP assay was performed according to kit instructions (Direct cAMP ELISA, Enzo Life Sciences).

*Statistical Analysis:* All graphs are presented as the mean ±SE of the results from independent preparations of cells (ie. N = 3–4 as indicated in the figure legends). Agonist treatments were compared to vehicle control performed on the same day and were added where indicated by the arrow. All data was analyzed by two-way unpaired ANOVA with Sidak's post-hoc test unless otherwise indicated. *$p<0.05$ **$p<0.001$ ***$p<0.0001$ ****$p<0.00001$ using GraphPad Prism 7.0.

## Acknowledgements

AVS Supported by NIH Grant R35GM127303; RI supported by NIH R00HL122508.

## Additional information

### Funding

| Funder | Grant reference number | Author |
| --- | --- | --- |
| National Institutes of Health | R35GM127303 | Alan V Smrcka |
| National Institutes of Health | R00HL122508 | Roshanak Irannejad |

The funders had no role in study design, data collection and interpretation, or the decision to submit the work for publication.

### Author contributions

Craig A Nash, Conceptualization, Data curation, Formal analysis, Investigation, Methodology, Writing—original draft; Wenhui Wei, Data curation, Formal analysis, Investigation, Writing—original draft, Writing—review and editing; Roshanak Irannejad, Resources, Writing—review and editing; Alan V Smrcka, Conceptualization, Formal analysis, Supervision, Funding acquisition, Project administration, Writing—review and editing

### Author ORCIDs

Alan V Smrcka https://orcid.org/0000-0003-3099-8812

### Ethics

Animal experimentation: This study was performed in strict accordance with the recommendations in the Guide for the Care and Use of Laboratory Animals of the National Institutes of Health. All of the animals were handled according to approved institutional animal care and use committee (IACUC) protocols of the University of Michigan protocol number PRO00009147.

### Decision letter and Author response

Decision letter https://doi.org/10.7554/eLife.48167.031
Author response https://doi.org/10.7554/eLife.48167.032

## Additional files

### Supplementary files
• Transparent reporting form
DOI: https://doi.org/10.7554/eLife.48167.029

### Data availability
Source data files have been provided.

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
