## [Decision Letter]

Thank you for submitting your article "Golgi localized β1-adrenergic receptors stimulate Golgi PI4P hydrolysis by PLCε to regulate cardiac hypertrophy" for consideration by *eLife*. Your article has been reviewed by three peer reviewers, one of whom is a member of our Board of Reviewing Editors, and the evaluation has been overseen by Vivek Malhotra as the Senior Editor. The reviewers have opted to remain anonymous.

The reviewers have discussed the reviews with one another and the Reviewing Editor has drafted this decision to help you prepare a revised submission.

The present study by Nash et al. is potentially of high relevance as it provides evidence that beta1-adrenergic receptors signal via Gs proteins on membranes of the Golgi complex in primary cardiomyocytes and is involved in catecholamine-induced hypertrophy. Moreover, they propose a novel mechanism whereby NE reaches (via Oct3) Golgi-resident beta1-adrenergic receptors to stimulate hypertrophic pathways through local activation of Epac/PLCepsilon/PKD. Overall, this is an interesting and well-conducted study on an emerging and highly relevant aspect of GPCR signaling.

The following are concerns that should be addressed in a significant revision before a final decision can be made.

1) Further work is needed to rule out the alternative interpretation that the signaling involves retrograde trafficking of internalized receptors/ligands to the trans-Golgi network/Golgi.

A) The lack of effect of the endocytosis inhibitor Dyngo is an important result and needs a positive to control to show Dyngo effectiveness. This could be done for instance by expressing fluorescently labelled beta1-adrenergic receptors and monitoring their internalization by fluorescence microscopy or, ideally, more quantitative approaches.

B) Show endogenous beta1-adrenergic receptors at the Golgi capable of activation of Gs (one should consider that showing mini-G recruitment to a certain subcellular compartment does not automatically demonstrate that endogenous G proteins are present and/or are activated by receptors in that compartment).

C) Confirm the Oct3 inhibitor does not block dobutamine activation of PI4P hydrolysis (similar to the control performed for hypertrophy). Additionally, it would be important to show that, as predicted by their model, beta1-adrenergic signaling at the plasma membrane (e.g. mini-G recruitment) is not affected by the Oct3 inhibitors.

D) Noticeably, the images and videos (although of relatively poor resolution) actually show rapid mini-G recruitment to what appears to be vesicular structures after agonist stimulation, strongly suggesting the presence of internalized receptors that are active in endosomes. Can the authors exclude that these internalized and active receptors in endosomes subsequently traffic to and accumulate in the trans-Golgi network to induce local Gs activation and PLCepsilon signaling? Could perhaps isoproterenol and dobutamine/NE induce dissimilar internalization/trafficking patterns, which might explain why only dobutamine/NE induce Golgi signaling?

2) Better establish the responsible signaling mechanisms at the Golgi:

A) Show adenylyl cyclase localization at the Golgi or that a specific pool of cAMP is induced by the Golgi beta1AR in response to NE.

B) Can RNAi (or CRISPR) experiments for Epac be performed to validate the link between PI4P hydrolysis by Golgi NE-activated beta1AR and the Epac/mAKAPβ/PLCε complex?

3) Improve the quantification/presentation of results:

In single cell experiments, each cell within a separate experiment is a replicate that contributes to the robustness of the mean for each repeat. The figure legends do not make clear how the authors are treating their data and whether they considered individual cells as separate "N" values for determination of variance/SE. This practice would be incorrect. The mean +/- SEM should be calculated from multiple independent experiments "N", not the cell number/experiments. The method or legend needs to be written in a way that makes it much clearer how statistics were done by indicating the number of independent experiments (N) and the number of cells per experiment (n).

4) Tone down conclusions regarding physiological relevance:

A) Given that this study has been performed in cultured cells, the statement "The work presented here elucidates a physiological function for these Golgi localized receptors" (Discussion) seems inadequate. The text should be modified/revised to clearly acknowledge the speculative nature of this with respect of Golgi beta1AR signaling in cardiac function.

B) Also at issue in this regard is the lack of direct evidence for the role endogenous receptor and Gs at the Golgi (as mentioned in 1B above), which raises questions regarding physiological significance.

5) Citation of previous work:

A) In the Introduction, where the authors introduce the new paradigm of signalling by GPCRs at intracellular sites, they should also cite the work on the TSH receptor by Calebiro et al., 2009, which was actually the first study, together with the one on the PTH receptor by JP Vilardaga's group, to show that GPCRs can signal via classical Gs-dependent pathways at intracellular sites.

B) Besides the recent findings on the beta1-adrenergic receptor, there is strong evidence that the TSH receptor signals in the Golgi/trans-Golgi network. This was already hypothesized based on trafficking and colocalization data in Calebiro et al., 2019 and then directly demonstrated in a recent study by the same group (Godbole et al., 2017). Interestingly, whereas in the case of the beta1-adrenergic receptor NA seems to reach Golgi-resident beta1-adrenergic receptors, in the case of the TSH receptor it has been demonstrated that this occurs via retrograde trafficking of receptors and hormones to the trans-Golgi network, where they activate a local pool of Gs, adenylyl cyclase and PKA II to induce efficient nuclear signalling. The authors should cite and discuss these recent findings on the TSH receptor and comment on the similarities/differences with the model they are proposing for the beta1-adrenergic receptor.

C) Adding references to the last paragraph of the Discussion section on Iso data would be helpful.

Other issues raised in the review that should be addressed.

6) The experiments raise the question of how norepinephrine gets from the cytosol into the Golgi lumen in the cell types studied. Is this required for its activation of receptors and, if so, is there a transporter expressed in these cells?

7) The signaling pathway with all its implicated components is reasonably complicated and should be presented in a schematic that shows their cellular location. This could be supplementary material. In particular, the Discussion leaves unclear the location of adenylyl cyclase that is activated by Golgi localized Gs.

8) Figures 1A and 4A: legends or figures do not make clear if ligands have been washout. It'd incorrect to state that NE or Dob mediates sustained PI4P hydrolysis without ligand washout.

9) Figure 1B, C: Specify whether these are merged fluorescent and brightfield images?

10) Figure 3A: GalT-mApple looks saturated and pixelated. In addition, images showing Nb80 in the Golgi after Rapamycin addition should be provided as a positive control of the experiments.

11) Figure 4B: these images and Figure 4—video 1 do not convincingly show recruitment of mGs-Venus at the plasma membrane and then to the Golgi. A better analysis should be provided. This reviewer recommends STED microscopy to demonstrate Golgi localization of the beta1AR/mGs complex in response to NE; calculating Pearson's correlation coefficients between mGs-venus and a Golgi fluorescent marker would also support authors' conclusion.

12) Figure 7A: Scale bars need to be included to show that images were acquired at the same magnification.

13) Figure 2: Western blot data showing the efficiency of the PLCε shRNA should be included.

14) The time point when agonists were added should be indicated in Figure 1D.

15) "si" needs to be corrected to "sh" in Figure 2A.

16) "NC" in Figure 2B needs to be defined in figure legend. Include RA1 viral transduction protocol in Materials and methods section.

17) Define "FOB" in figure legends for Figures 7B and 8.

18) Figure 7B: vehicle and Metoprolol images have what appears to be bubbles on top of the cells. These artefacts usually affect image analysis. Were these images used for the data provided?

---

## [Author Response]

[…] The following are concerns that should be addressed in a significant revision before a final decision can be made.1) Further work is needed to rule out the alternative interpretation that the signaling involves retrograde trafficking of internalized receptors/ligands to the trans-Golgi network/Golgi.A) The lack of effect of the endocytosis inhibitor Dyngo is an important result and needs a positive to control to show Dyngo effectiveness. This could be done for instance by expressing fluorescently labelled beta1-adrenergic receptors and monitoring their internalization by fluorescence microscopy or, ideally, more quantitative approaches.

This data is now included in Figure 5—figure supplement 1. We used N-terminally FLAG tagged β1AR and labeled intact cells with anti-FLAG M2 antibody conjugated to Cy3. All of the labeling was at the cell surface. Dobutamine or NE addition caused weak internalization consistent with previously published results showing that β1ARs do not internalize well (1, 2). No Golgi labeling was observed. Strong stimulation with Iso did cause internalization that was blocked by pretreatment with Dyngo providing a positive control for the efficacy of Dyngo supporting the data shown in Figure 5. Together these data strongly suggest that internalization is not responsible for activation that is observed at the Golgi by these two agonists as had previously been shown (1).

B) Show endogenous beta1-adrenergic receptors at the Golgi capable of activation of Gs (one should consider that showing mini-G recruitment to a certain subcellular compartment does not automatically demonstrate that endogenous G proteins are present and/or are activated by receptors in that compartment).

We agree completely. The Mini-G experiments are presented to bolster the majority of the data with endogenous receptors G proteins and signaling components. Significantly, the key experiment using Golgi targeted NB80 to block the endogenous signaling pathway is strong evidence that the endogenous β1ARs are signaling at the Golgi apparatus (1). We have tried to observe recruitment of Mini-Gs with endogenous receptors and have not been able to get convincing images likely due to the relatively low levels of receptor present, that nevertheless are quite capable of stimulating the endogenous signaling cascades. We were able to detect endogenous β1ARs in cardiac myocyte now shown in Figure 1—figure supplement 1. Other work has clearly demonstrated the presence of G proteins at the Golgi apparatus (3-5). Thus our conclusions are entirely consistent with the literature.

C) Confirm the Oct3 inhibitor does not block dobutamine activation of PI4P hydrolysis (similar to the control performed for hypertrophy).

We now include in Figure 5—figure supplement 1C, data showing that Oct3 inhibition does not block dobutamine-dependent PI4P hydrolysis. Surprisingly we see an enhancement of dob-dependent PI4P hydrolysis in the presence of corticosterone which we do not understand at this point.

Additionally, it would be important to show that, as predicted by their model, beta1-adrenergic signaling at the plasma membrane (e.g. mini-G recruitment) is not affected by the Oct3 inhibitors.

We now include data in Figure 5—figure supplement 1B that Oct3 inhibition does not affect Iso dependent cAMP production indicating that Oct3 inhibition does not affect β1AR signaling at the plasma membrane.

D) Noticeably, the images and videos (although of relatively poor resolution) actually show rapid mini-G recruitment to what appears to be vesicular structures after agonist stimulation, strongly suggesting the presence of internalized receptors that are active in endosomes. Can the authors exclude that these internalized and active receptors in endosomes subsequently traffic to and accumulate in the trans-Golgi network to induce local Gs activation and PLCepsilon signaling? Could perhaps isoproterenol and dobutamine/NE induce dissimilar internalization/trafficking patterns, which might explain why only dobutamine/NE induce Golgi signaling?

Please see our response to point 1A. We do not believe these structures are endosomes but rather may be Gs on microtubules. There are multiple reports that Gs binds to tubulin (6, 7). It is not clear whether they are recruited to these structures due to receptor activation or whether they just become visible when the cytoplasm is cleared of the mini-G fluorescence upon receptor activation. They appear with much faster kinetics than does movement of β1ARs into endosomes with Iso stimulation and once they appear they do not traffic but rather are stationary. Additionally, as discussed in point 1A above, NE and dobutamine cause weak internalization of the βARs, and when we label cell surface receptors with anti-FLAG antibody none of this fluorescence appears in the Golgi when treated with any agonist (Figure 5—figure supplement 1A). Finally, the results with Dyngo showing that it does not affect NE-dependent PI4P hydrolysis (Figure 5D and E), coupled with the new positive control data in Figure 5—figure supplement 1A, confirms that the signaling at the Golgi is not dependent on receptor internalization into endosomes.

2) Better establish the responsible signaling mechanisms at the Golgi:A) Show adenylyl cyclase localization at the Golgi or that a specific pool of cAMP is induced by the Golgi beta1AR in response to NE.

Showing adenylyl cyclase localization at the Golgi is likely not possible due to the low abundance of the native protein and poor antibodies. To my knowledge no one has successfully localized native adenylyl cyclase isoforms in cells. On the other hand it has previously been shown that adenylyl cyclase binds to mAKAPβ in cardiac myocytes (8). Thus, there is likely a local adenylyl cyclase that can be accessed by Gs in the Golgi compartment.

With regard to visualization of a specific pool of cAMP at the golgi: Many others have shown specific compartmentation of cAMP in cells including in cardiac myocytes and are perhaps more adept at these analyses than we are (for example: (9). cAMP compartmentation is a well-established biological phenomenon. I would also argue that just because you can see cAMP in a particular compartment, this does not define a functional role, and if you can’t see it, it does not mean there is none generated due to possible limitations of FRET sensors. We believe that our contribution is to ascribe a functional role for a specific cAMP pool and we believe the body of evidence we present is sufficient to support the conclusions of this work.

B) Can RNAi (or CRISPR) experiments for Epac be performed to validate the link between PI4P hydrolysis by Golgi NE-activated beta1AR and the Epac/mAKAPβ/PLCε complex?

We could do this but we have already provided extensive evidence in multiple publications that Epac at mAKAPβ is involved in cAMP dependent regulation of PI4P hydrolysis (10-12). The compound we use in this study, HJC, is a highly specific inhibitor of Epac (13). Thus we feel this link is already well established. The primary contribution of this manuscript is to show that Golgi localized receptors are required for cAMP to access Epac in this complex.

3) Improve the quantification/presentation of results:In single cell experiments, each cell within a separate experiment is a replicate that contributes to the robustness of the mean for each repeat. The figure legends do not make clear how the authors are treating their data and whether they considered individual cells as separate "N" values for determination of variance/SE. This practice would be incorrect. The mean +/- SEM should be calculated from multiple independent experiments "N", not the cell number/experiments. The method or legend needs to be written in a way that makes it much clearer how statistics were done by indicating the number of independent experiments (N) and the number of cells per experiment (n).

All of the graphs have been remade where N is represents the number of biological replicates from individual experiments with cells isolated from individual animals. These experiments are all done with primary myocytes which requires preparations from separate animals for each experiment since myocytes cannot be passaged in culture.

4) Tone down conclusions regarding physiological relevance:A) Given that this study has been performed in cultured cells, the statement "The work presented here elucidates a physiological function for these Golgi localized receptors" (Discussion) seems inadequate. The text should be modified/revised to clearly acknowledge the speculative nature of this with respect of Golgi beta1AR signaling in cardiac function.

I have done this somewhat by changing the wording in various places to say “possible physiological functions”. These are primary cardiac myocytes acutely isolated from neonatal rats. This greatly raises the difficulty of these experiments, with the goal that studies from these physiologically relevant cells have significant relevance to physiology. Neonatal myocytes beat in culture and respond to hypertrophic stimuli with increases in cell size, hypertrophic gene expression, and protein synthesis. We also show that this system operates in adult myocytes. Neonatal myocytes while not always perfect are a well established model for studying cardiac hypertrophy. I agree that these experiments will have to be backed up in an animal model of heart failure, but they should be given credit for having some meaningful level of physiological relevance.

B) Also at issue in this regard is the lack of direct evidence for the role endogenous receptor and Gs at the Golgi (as mentioned in 1B above), which raises questions regarding physiological significance.

Except for the experiments with Venus-Mini-Gs, which were done with transfected β1AR, *all* of the experiments were done with the endogenous signaling system present in cardiac myocytes. The role of endogenous internal receptors in general was clearly established by multiple methods. The key experiments demonstrating a role for endogenous receptors at the Golgi are shown in Figure 3B and Figure 5F where targeting of a nanobody, Nb80, to the Golgi apparatus blocks the ability of dobutamine or NE to stimulate PI4P hydrolysis, and in Figure 8C and D, blocks cardiomyocyte hypertrophy. This nanobody specifically binds to βARs and competes for βAR-G protein interactions, thus demonstrating that endogenous Golgi localized βAR-dependent G protein activation is responsible for stimulation of PI4P hydrolysis and hypertrophy. Finally, this is consistent with previously published work in Hela cells showing endogenous β1AR signaling at the Golgi (1)

5) Citation of previous work:A) In the Introduction, where the authors introduce the new paradigm of signalling by GPCRs at intracellular sites, they should also cite the work on the TSH receptor by Calebiro et al., 2009, which was actually the first study, together with the one on the PTH receptor by JP Vilardaga's group, to show that GPCRs can signal via classical Gs-dependent pathways at intracellular sites.

This reference was added to the last paragraph of the Introduction.

B) Besides the recent findings on the beta1-adrenergic receptor, there is strong evidence that the TSH receptor signals in the Golgi/trans-Golgi network. This was already hypothesized based on trafficking and colocalization data in Calebiro et al., 2019 and then directly demonstrated in a recent study by the same group (Godbole et al., 2017). Interestingly, whereas in the case of the beta1-adrenergic receptor NA seems to reach Golgi-resident beta1-adrenergic receptors, in the case of the TSH receptor it has been demonstrated that this occurs via retrograde trafficking of receptors and hormones to the trans-Golgi network, where they activate a local pool of Gs, adenylyl cyclase and PKA II to induce efficient nuclear signalling. The authors should cite and discuss these recent findings on the TSH receptor and comment on the similarities/differences with the model they are proposing for the beta1-adrenergic receptor.

A discussion of these concepts has been added to the subsection “Compartmentalized GPCR signaling”.

C) Adding references to the last paragraph of the Discussion section on Iso data would be helpful.

Some references have been added to this section.

Other issues raised in the review that should be addressed.6) The experiments raise the question of how norepinephrine gets from the cytosol into the Golgi lumen in the cell types studied. Is this required for its activation of receptors and, if so, is there a transporter expressed in these cells?

Yes the Oct3 transporter has been shown to be present on the plasma membrane AND internal membranes in cardiac cells (14) and other cells (15) and is one possible mechanism for transport into the Golgi lumen. This is now diagrammed in Figure 9.

7) The signaling pathway with all its implicated components is reasonably complicated and should be presented in a schematic that shows their cellular location. This could be supplementary material. In particular, the Discussion leaves unclear the location of adenylyl cyclase that is activated by Golgi localized Gs.

A new figure (Figure 9) now has a schematic showing the various components of the system. This is now discussed in the second paragraph of the subsection “Compartmentalized GPCR signaling”.

8) Figures 1A and 4A: legends or figures do not make clear if ligands have been washout. It'd incorrect to state that NE or Dob mediates sustained PI4P hydrolysis without ligand washout.

The ligands have not been washed out. I’m not sure why this is incorrect. In the presence of agonist the response is sustained.

9) Figure 1B, C: Specify whether these are merged fluorescent and brightfield images?

These are separate mini-Gs-Venus and CFP golgi images that have not been merged. This has been clarified in the figure legend.

10) Figure 3A: GalT-mApple looks saturated and pixelated. In addition, images showing Nb80 in the Golgi after Rapamycin addition should be provided as a positive control of the experiments.

A new image has been inserted in place of Figure 3A demonstrating recruitment of Nb80-GFP to the golgi in response to rapamycin.

11) Figure 4B: these images and Figure 4—video 1 do not convincingly show recruitment of mGs-Venus at the plasma membrane and then to the Golgi. A better analysis should be provided. This reviewer recommends STED microscopy to demonstrate Golgi localization of the beta1AR/mGs complex in response to NE; calculating Pearson's correlation coefficients between mGs-venus and a Golgi fluorescent marker would also support authors' conclusion.

We calculated a pearson’s coefficient from multiple images and this reported in the figure legend for 4B.

12) Figure 7A: Scale bars need to be included to show that images were acquired at the same magnification.

Scale bars have been added.

13) Figure 2: Western blot data showing the efficiency of the PLCε shRNA should be included.

A western blot has been added to Figure 2.

14) The time point when agonists were added should be indicated in Figure 1D.

This has been added to the figure.

15) "si" needs to be corrected to "sh" in Figure 2A.

This has been done.

16) "NC" in Figure 2B needs to be defined in figure legend. Include RA1 viral transduction protocol in Materials and methods section.

These have been done.

17) Define "FOB" in figure legends for Figures 7B and 8.

This has been done.

18) Figure 7B: vehicle and Metoprolol images have what appears to be bubbles on top of the cells. These artefacts usually affect image analysis. Were these images used for the data provided?

Yes these images were used to quantitate the data provided but the out of focus “bubbles” are only in the DAPI channel which was not used for the quantitation.

1) R. Irannejad et al., Functional selectivity of GPCR-directed drug action through location bias. Nat. Chem. Biol. 13, 799-806 (2017).

2) T. Shiina, A. Kawasaki, T. Nagao, H. Kurose, Interaction with β-Arrestin Determines the Difference in Internalization Behavior between β1- and β2-Adrenergic Receptors. J. Biol. Chem. 275, 29082-29090 (2000).

3) P. Melancon et al., Involvement of GTP-binding "G" proteins in transport through the Golgi stack. Cell 51, 1053-1062 (1987).

4) C. Jamora et al., Gbg-mediated regulation of Golgi organization is through the direct activation of protein kinase D. Cell 98, 59-68 (1999).

5) R. Irannejad, P. B. Wedegaertner, Regulation of Constitutive Cargo Transport from the trans-Golgi Network to Plasma Membrane by Golgi-localized G Protein βγ Subunits. J. Biol. Chem. 285, 32393-32404 (2010).

6) N. Wang, K. Yan, M. M. Rasenick, Tubulin binds specifically to the signal-transducing proteins, Gs α and Gi α 1. J. Biol. Chem. 265, 1239-1242 (1990).

7) T. Sarma et al., Activation of Microtubule Dynamics Increases Neuronal Growth via the Nerve Growth Factor (NGF)- and Gαs-mediated Signaling Pathways. J. Biol. Chem. 290, 10045-10056 (2015).

8) M. S. Kapiloff et al., An Adenylyl Cyclase-mAKAPb Signaling Complex Regulates cAMP Levels in Cardiac Myocytes. J. Biol. Chem. 284, 23540-23546 (2009).

9) N. C. Surdo et al., FRET biosensor uncovers cAMP nano-domains at β-adrenergic targets that dictate precise tuning of cardiac contractility. Nature Communications 8, 15031 (2017).

10) L. Zhang et al., Phospholipase Cε hydrolyzes perinuclear phosphatidylinositol 4-phosphate to regulate cardiac hypertrophy. Cell 153, 216-227 (2013).

11) C. S. Brand, R. Sadana, S. Malik, A. V. Smrcka, C. W. Dessauer, Adenylyl Cyclase 5 Regulation by Gβγ Involves Isoform-Specific Use of Multiple Interaction Sites. Mol. Pharmacol. 88, 758-767 (2015).

12) C. A. Nash, L. M. Brown, S. Malik, X. Cheng, A. V. Smrcka, Compartmentalized cyclic nucleotides have opposing effects on regulation of hypertrophic phospholipase Cepsilon signaling in cardiac myocytes. J Mol Cell Cardiol 121, 51-59 (2018).

13) Y. Zhu et al., Biochemical and pharmacological characterizations of ESI-09 based EPAC inhibitors: defining the ESI-09 "therapeutic window". Sci Rep 5, 9344 (2015).

14) C. D. Wright et al., Nuclear α1-Adrenergic Receptors Signal Activated ERK Localization to Caveolae in Adult Cardiac Myocytes. Circ Res 103, 992-1000 (2008).

15) M. Roth, A. Obaidat, B. Hagenbuch, OATPs, OATs and OCTs: the organic anion and cation transporters of the SLCO and SLC22A gene superfamilies. Br. J. Pharmacol. 165, 1260-1287 (2012).